# A model of thalamo-cortical interaction for incremental binding in mental contour-tracing

**Daniel Schmid**[1]*, **Heiko Neumann**[1]

Institute for Neural Information Processing, Ulm University, Ulm, Baden-Württemberg, Germany

* daniel-1.schmid@uni-ulm.de

## Abstract

Object-basd visual attention marks a key process of mammalian perception. By which mechanisms this process is implemented and how it can be interacted with by means of attentional control is not completely understood yet. Incremental binding is a mechanism required in demanding scenarios of object-based attention and is experimentally well investigated. Attention spreads across a representation of the visual object and labels bound elements by constant up-modulation of neural activity. The speed of incremental binding was found to be dependent on the spatial arrangement of distracting elements in the scene and to be scale invariant giving rise to the growth-cone hypothesis. In this work, we propose a neural dynamical model of incremental binding that provides a mechanistic account for these findings. Through simulations, we investigate the model properties and demonstrate how an attentional spreading mechanism tags neurons that participate in the object binding process. They utilize Gestalt properties and eventually show growth-cone characteristics labeling perceptual items by delayed activity enhancement of neuronal firing rates. We discuss the algorithmic process underlying incremental binding and relate it to our model computations. This theoretical investigation encompasses complexity considerations and finds the model to be not only of explanatory value in terms of neurophysiological evidence, but also to be an efficient implementation of incremental binding striving to establish a normative account. By relating the connectivity motifs of the model to neuroanatomical evidence, we suggest thalamo-cortical interactions to be a likely candidate for the flexible and efficient realization suggested by the model. There, pyramidal cells are proposed to serve as the processors of incremental grouping information. Local bottom-up evidence about stimulus features is integrated via basal dendritic sites. It is combined with an apical signal consisting of contextual grouping information which is gated by attentional task-relevance selection mediated via higher-order thalamic representations.

## Author summary

Understanding a visual scene requires us to tell apart visual objects from one another. Object-based attention is the process by which mammals achieve this. Mental processing

**Data availability statement:** All code for the model, creating simulation data, running

experiments, and plotting is available via a GitHub repository (https://github.com/schmidDan/incremental_binding).

**Funding:** The authors received no specific funding for this work.

**Competing interests:** The authors have declared that no competing interests exist.

of object components determines whether they are compatible to the overall object and, thus, should be grouped together to be perceived as a whole or not. For complex objects exhibiting great variability in the spatial arrangement of feature compositions, or for complex, ambiguous scenes, this processing needs to progress serially, determining the compatibility step by step. In this work, we propose a neural model of such processes and try to answer the question of how it might be implemented in the brain. We test the model on a case of object-based attention for grouping elongated lines and compare it to the available experimental evidence. We additionally show that the model not only explains this evidence, but it does so also by making efficient use of neurons and connections—a property likewise desirable for brains and machines. Together, these findings suggest which brain areas might be involved in realizing object-based attention and how to reason about the complexity of this computation.

## 1. Introduction

Visual cortex is recruited for a variety of tasks, such as searching a scene for the presence of an object, visual tracking of moving objects, or making visual comparison judgments. Many of these tasks are based on seemingly effortless perception that relies on processes operating in parallel. The visual system can employ different processing strategies during different temporal phases of processing [1]. Object recognition can work rapidly, oftentimes relying on detecting learned patterns during a single sweep of feedforward propagation followed by automatic parallel grouping schemes establishing relations between these patterns [2–4]. Recognizing highly ambiguous objects requires the dedicated deployment of functional resources and only can be executed sequentially [5–7]. Such deployment of functional resources is controlled by selective attention. Endogenous attentional control steers the sequence of visual mental operations. Each sequence forms a dedicated pattern of neural activity and can be used to establish a specific visual function [8–10]. Object-based attention is one such cognitive function, which requires grouping together visual components that may belong to the same object. The goal is to segregate these from distracting parts of a scene, to assemble a visual object, and to disambiguate between possible alternatives of explanation [11].

To achieve this goal, two forms of grouping, or binding, can be distinguished: parallel and incremental grouping [8,9]. *Parallel grouping* automatically extracts matching input patterns into representational elements and groups them by a fixed scheme that determines compatibility based on statistics of co-occurrence. Ambiguities that cannot be resolved by parallel grouping require volitional incremental grouping processes [5,9]. *Incremental grouping* iteratively and dynamically evaluates compatibility of previously grouped elements with their neighbors [12–14]. If compatible neighbors exist, they become part of the grouped elements as well and the iterative process likewise continues for their neighbors to discern the assembled shape [15].

Executing an incremental grouping operation serves to bind visual elements through the evaluation of spatial relations that are not hard-wired as in automatic grouping operations. For example, incremental grouping is deployed to decide upon a spatial relation of two disparate locations in the visual scene. In contour tracing tasks subjects use incremental grouping to determine the end point of a connected line while maintaining fixation of a reference location ([16]; Fig 1). Execution of contour tracing requires four phases of processing [20]. During the first phase, features are detected by an onset response. The second phase establishes a *base representation* of foreground and background elements by parallel grouping. In

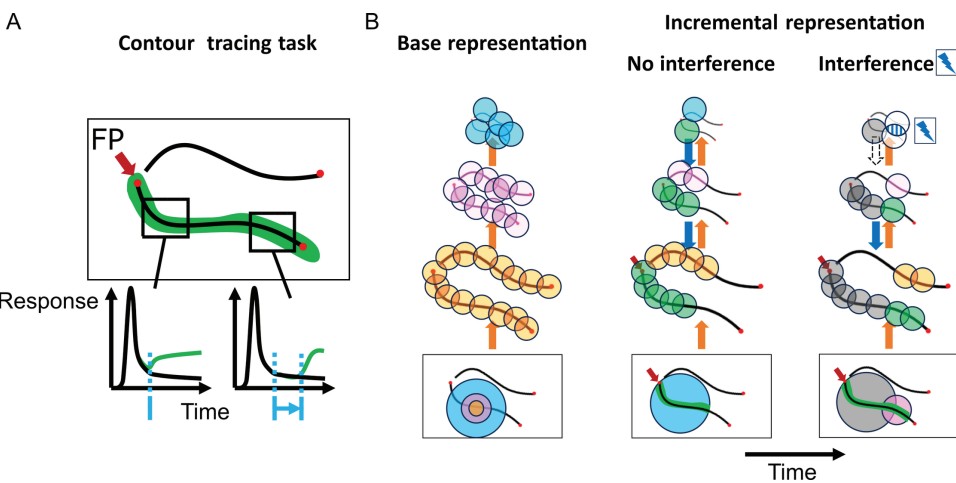

**Fig 1. Incremental binding in contour tracing tasks and the growth cone hypothesis.** (**A**) During contour tracing tasks a fixation point (FP) is presented together with a pair of contours to non-human primate subjects. Subjects need to determine which contour is connected with the fixation point and perform a saccade towards the contour's end point in response. Neurons coding for the connected contour show an up-modulation of their activity (green) with respect to a baseline response profile (black) [16]. The onset latency of this up-modulation is thereby dependent on the distance along the line from the fixation point (blue). Furthermore, the up-modulation is found to be persistent over time, so that ultimately the whole contour representation is up-modulated. These findings indicate that incremental binding utilizes a serial process of grouping relatable items after an initial phase of stimulus response. (**B**) A specific hypothesis of incremental binding is the growth-cone hypothesis [14,17]. It explains how speed-variations in incremental binding arise from the distance to distracting elements in the vicinity of the target object. The initial processing phase has extracted features of the stimulus in parallel to build a hierarchical base representation (left). Within this hierarchy incremental binding starts from an attentional seed position (red arrow) and can proceed at different scales with representations evolving over time (center, right). While coarser scales span more space and can proceed faster ("no interference" case, center), they might suffer from interference with nearby elements. In the case of interference finer scales still can perform incremental binding, but with a reduced speed ("interference" case, right; hatched area at top-most scale and missing feedback arrow). Neural receptive fields are represented by colored circles as an overlay to the input image and are drawn with their relative sizes. Colors are matching the respective level of the hierarchy on which the neurons are localized. Green colored circles denote neurons that are up-modulated and have become part of the incremental representation. Gray colored circles on the target curve denote neurons already up-modulated during the previous timestep. For the same amount of time, more neurons can be grouped in case higher, coarser scales can participate in incremental binding. Colored circles on the distractor curve are only displayed if relevant for the current interaction. The red arrow in the input displays denotes the location of the seed of attention, which triggers the execution of the incremental binding process.

contour tracing scenarios this base representation encodes bottom-up evidence for the existence of line elements that match the respective neuron's retinotopic position and feature selectivity. During the third phase, an *incremental representation* is computed by combining the existing evidence with an attentional binding signal into elements of a perceptual object. After the process has converged, during a fourth phase, a decision is made based on the integrated evidence about the visual objects. The formation of the incremental representation during the third phase relies on incremental grouping to propagate the binding signal and iteratively group the attended object [15]. The binding signal consists of two kinds of information: contextual information and task-related information. *Contextual information* stems from a neuron's spatial neighborhood and serves to evaluate *compatibility* between the features a neuron and its neighbors encode. *Task-related information* signals whether this compatibility is behaviorally relevant and provides the attentional seed point for building the incremental representation. This seed is selected, or indexed, by the fixation point in contour tracing panels.

In order to build such incremental representations different neural coding schemes have been proposed [5,11]. To make the presence of an object explicit, one candidate scheme instantiates all possible combinations through hard-coded feature selections. Such conjunctive code requires an expensive realization in terms of the number of neurons and their connections. Alternatively, binding through synchronization of neural oscillations and the delayed modulation of neural firing rates have been proposed as other schemes [21]. The latter frameworks are more flexible and more efficient to implement, particularly utilizing those neurons which are already involved in the base representation. Such neurons can be recruited dynamically and context-dependently as incremental components to form a perceptual object. The evidence from neural correlates about the incremental binding mechanism indicates a constant up-modulation of neural firing rates across the whole attended object [11,16]. This further discerns the spreading of object-based attention by incremental binding from the idea of an attentional spotlight, where only a proportion of the object would be modulated at a time [22]. Additionally, the onset time of the modulation depends on the neuron's distance along the curve [16]. This is taken as evidence that an active labeling process operates to incrementally propagate the binding signal along the object.

Does such contour labeling operate independently of the context of a scene? Further investigations showed that the speed of incremental binding is dependent on the target object's distance to distracting neighboring elements and that this behavior is best explained in terms of the so-called *growth-cone hypothesis* [14,17]. The growth-cone hypothesis incorporates the observation that incremental binding is a scale-invariant process [14,17,23]. The local speed of incremental binding is determined by the neuron with the largest receptive field, which does not interfere with distracting elements in the scene. This way, incremental binding speed is high if the relative distance to distractors is large. If distractors are located close by, binding signals are propagated by neurons with smaller receptive fields. This reduces interference but leads to slower propagation speeds (Fig 1).

Further evidence has been accumulated about incremental grouping in recent studies. These can be categorized by the kind of species in which they had been conducted, by the kind of correlates that have been found, and by the kind of stimuli that were used. Notably, studies so far only investigated the phenomenon in (non-)human primates but not other mammals. Neural correlates have mostly been measured in non-human primates, with one exception stemming from a human epilepsy patient [24]. There, experiments are either based on variants of the contour tracing tasks, where the to-be-grouped object elements are lines [16,17,25–33], or are based on abstract shapes where the to-be-grouped elements are surface regions of the objects [20,34]. Human experiments were mostly restricted to reaction time measurements, of which many have been conducted based on curve displays [5,12,23,35–39] and some based on shapes and semantic objects [14,40] or real-world images [41,42]. Overall, incremental grouping phenomena can be found at least in human and non-human primates and for a broad set of contour- and region-based visual stimuli alike. This broad range of findings highlights incremental binding as an essential mechanism for visual-cognitive processes of object-based attention.

Aside from experimental studies, incremental binding has been investigated on all levels of analysis [18]. On an *implementational level* models were proposed, of which some are mechanistic and some are rather phenomenological. The different models address different neuroanatomical scales and range from a laminar circuit level [43], over the level of local pools of neurons representing the computation by predefined interactions [44–46], to the level of neural networks exhibiting learned interactions to perform incremental binding tasks [47–49], to match behavioral evidence [42,50], or to explain object-based binding by a diffusion-like labeling mechanism [51]. On an *algorithmic level* more abstract descriptions

either discuss incremental binding directly [8,13,15,52] or its embedding within cognitive models [10]. Likewise, the *computational level* of incremental binding is discussed in terms of visual routines and cognitive programs [8–10] and provides connections to theories about binding in general, which embed the computation within the broader context of object-based visual processing [11,53–59]. Yet, integrative models that link across the levels of analysis are scarce. This current state of findings motivates the following questions: *Are there generic principles for mechanisms of incremental binding that also help to better understand and compare existing models? Is there a normative account that can guide the choice of a minimal set of elements to implement such mechanism? And, what predictions about binding in the brain would arise from a model which adheres to such generic and normative principles?*

In this work, we present a mechanistic model for incremental binding that spans the multiple levels of explanation [18] and links them to a normative account of connection and representational complexity (Fig 2). On an *implementational level*, it proposes how local pyramidal cell computation and inter-areal interactions can implement the incremental grouping algorithm while utilizing only a minimalistic set of computational elements and connectivity patterns. Through extensive simulations, we show that the model explains a broad range of experimental evidence. On the *algorithmic level*, we analyze the model within a graph-based model framework of incremental binding. We consider the model's functional processing and provide complexity estimates for different realizations of the incremental grouping algorithm. In this more functional investigation, we consider several architectural principles of distributed neural processing that could implement the desired computational operations for incremental binding. Furthermore, the serial composition of multiple elemental operations into a visual routine defines additional constraints. We argue that the wiring and, thus, representational network complexity, can be utilized to assess the efficacy of a specific network structure and its combination of components. We show that the network architecture proposed in this contribution possesses favorable properties regarding the complexity of its connectivity and neural code. On the *computational level*, we suggest that respecting these complexity considerations yields a normative account of incremental grouping in object-based attention. Considering the simulation results, the model architecture, and recent experimental evidence about functional anatomy, we then suggest coupled cortico-cortical and cortico-thalamo-cortical loops as a likely target circuitry that could implement such incremental binding mechanism in the brain and provide testable model predictions.

## 2. Results

Based on the motivations above, we now present the mechanistic model architecture that implements the neural incremental binding operation. Through numerical simulations, it explains a broad range of experimental evidence using only a single set of model parameters. An additional theoretical consideration relates the model to other possible architectural designs of incremental binding and reveals its favorable properties regarding model complexity.

### 2.1. The incremental binding model

The resulting model implements experimental findings about incremental binding and offers a mechanistic explanation by means of a dynamical neural network architecture (Fig 3). The model is described at a mesoscopic level, where neural processors resemble the computation of a local neural population. The neural processors capture functional principles of pyramidal cell computation and utilize a rate-based encoding scheme. The architecture is organized into

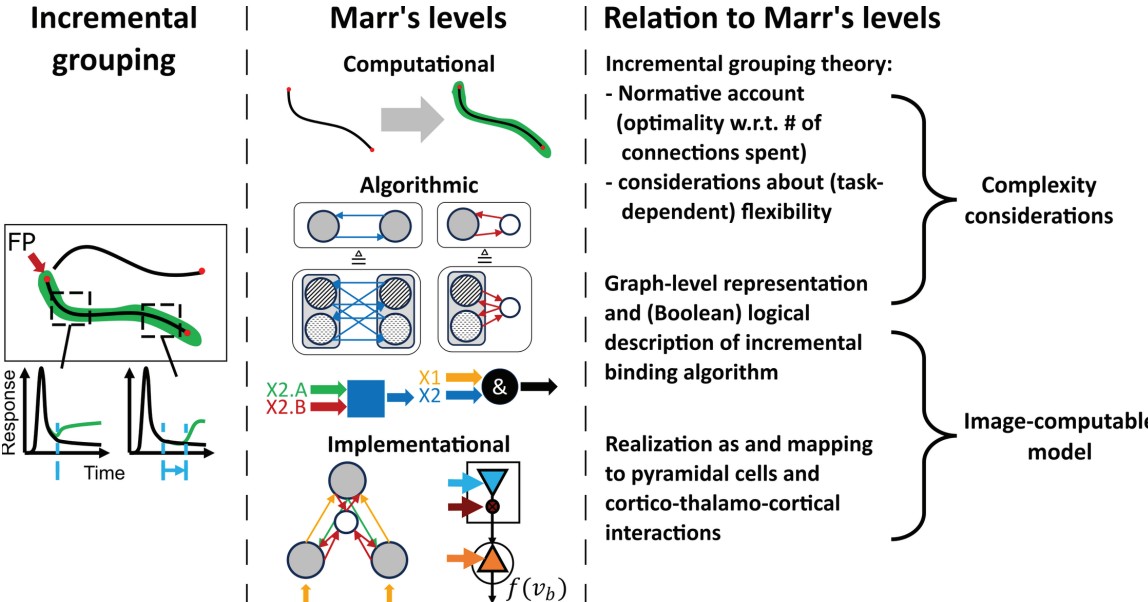

**Fig 2. The proposed model leverages different levels of analysis.** The presented study provides insights into incremental grouping among Marr's three levels of analysis [18]. On the computational level a normative account is provided for efficient connectivity and computation of incremental grouping in object-based attention. The normative account is based on complexity considerations, that are grounded in an algorithmic graph-based perspective of the incremental binding algorithm [15] and that are inspired by earlier work on computational complexity of vision processing [19]. On the implementational level we provide an image-computable dynamical neural model that respects these complexity considerations and propose how a biological network of pyramidal cells and higher-order thalamic neurons could form interacting cortico-cortical and cortico-thalamo-cortical loops to implement this model.

modules with distinct functions. It consists of a visual cortex module, an interfacing module, and a task module. The *visual cortex module* is organized in a hierarchy of retinotopic feature maps, which represent oriented contrast information in a scale-space pyramid. Feature maps order neurons in two-dimensional sheets. There, neurons are placed at retinotopic locations in a rectangular grid, code for specific stimulus features and exhibit local connectivity patterns. The visual cortex module computes the base representation as well as the incremental representation and propagates information among feed-forward and feedback connections from lower levels to higher levels and vice versa. Feedforward filtering is based on more localized information and extracts features for the next higher levels, while feedback filtering is based on larger regions and provides contextual information to stages at lower levels (see Methods for details). Overall, receptive field sizes grow along the stages of the hierarchy in accordance with neural data [61]. The *interfacing module* is likewise composed of neurons in a two-dimensional sheet. Yet, different from the visual cortex module, the neurons do not encode multiple stimulus features, but only whether a retinotopic position is part of the attended object or not. The module serves to interface task-related information between the task module and the different hierarchical levels of the visual cortex module, and broadcasts information about the binding state originating from one of the hierarchical levels to all other ones. Due to its connectivity, it forms a shallow, rather than a hierarchical interaction with the visual cortex module and is, thus, associated with higher-order (HO) thalamic regions [62]. The *task module* represents an attentional seed location. By this seed location the task module selects the starting position of the incremental binding operation. This selection happens by forwarding the location to the interfacing module. There, it constitutes the first entry in

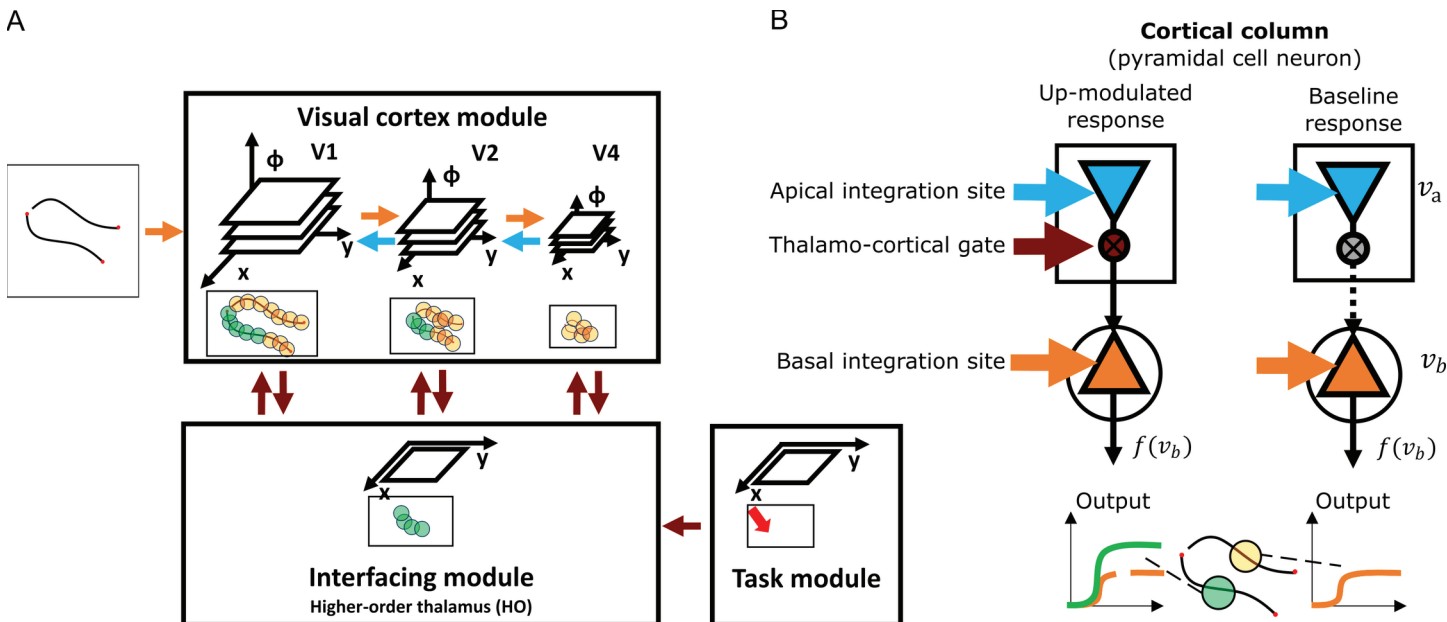

**Fig 3. Thalamo-cortical model architecture for incremental binding with local grouping processors that are represented by cortical columnar pyramidal cell computation.** (**A**) The proposed incremental binding architecture consists of three modules. The visual cortex module encodes the stimulus in a high-dimensional feature space organized along a spatial hierarchy. Adjacent hierarchical levels interact recurrently via feedforward (orange) and feedback (blue) connections. The interfacing module encodes task-relevant spatial locations. It is reciprocally connected with the visual cortex module aggregating information per location across features and projecting information back onto a local neighborhood irrespective of feature expression and hierarchical scale. The task module provides external input to the interfacing module to set the initial seed point of attention. It is proposed that these modules have biological counterparts in visual cortex areas V1 to V4 (visual cortex module), higher-order visual thalamus (interfacing module) and regions involved in attentional selection in either prefrontal, parietal, or temporal cortical areas (task module; the red arrow denotes the location for attentional selection). (**B**) Neural computation of cortical columns within the visual cortex module is represented by dynamical, rate-based pyramidal cell models. Each model pyramidal cell consists of two compartments and three types of input. The two compartments describe basal and apical integration sites of pyramidal cells [60]. From the coincidence of the three types of inputs local grouping information is evaluated to change the neural activity from a base representation state into an up-modulated activity state coding for binding information. There, feedforward signals about the base representation arrive at the basal compartment and can drive the cell into a baseline level activity (orange), while feedback signals arrive apically and are combined with thalamo-cortical gating signals before having an up-modulating effect on the basal computation. This up-modulated state of activity represents the binding state (green).

the sheet and, thus, determines the starting point from where an object should be incrementally grouped by attention. Such selection process is necessary to decide which object should be grouped and can likewise be useful in disambiguating the representation.

Within the visual cortex module, each retinotopic position is represented by a *model pyramidal cell neuron*. These model neurons abstractly capture the computation of a cortical column. Each neuron engages in the computation of the base representation and the incremental representation. Model neurons possess two sites of integration. These describe a basal dendritic site and an apical site of integration [60]. At the *basal site* input-related information enters via feed-forward signal propagation and is required to compute the base representation. This base representation results in a neuronal activity level that encodes the likelihood of feature presence. At the *apical site* contextual information enters via feedback signals propagated from higher levels and is integrated with binding signal information from the interfacing module. In case both kinds of information coincide, the apical site relays this signal to the basal site. There, it can be integrated with the base representation to compute the incremental representation. Thus, the incremental representation depends on the co-incidence of basal and apical signals of a model neuron. As a result, the neuron's activity level will be up-modulated and shifted towards a regime of higher activity. Such up-modulated activity

distributions represent the labeling of neural elements to signal binding through enhanced firing rates. The interfacing module aggregates firing rates from the pyramidal cell neurons of the visual cortex module. It filters these rates for such higher activity level to represent the binding state at that retinotopic position. The label is then broadcast back to spatial neighbors across scales in the visual cortex module to further propagate the binding signal locally.

## 2.2. Simulation results capture experimental evidence about incremental binding

### 2.2.1. Tracing time depends on contour length and captures neural correlates.
To investigate the incremental binding capabilities of the model we evaluated it on contour tracing tasks similar to those experiments conducted with human and non-human primate subjects (Fig 4). The model is able to capture the main findings and neural correlates. The model neurons encode the binding state by a constant up-modulation of their firing rate. The incremental nature of this binding process is visible in the time course of this up-modulation. After an initial onset phase, neurons located along the attended curve become up-modulated. There, the time when modulation begins depends on the neuron's distance along the contour from the seed point of attention. Once all neurons along the contour have been bound, the representation reaches a stable equilibrium, and the incremental binding mechanism comes to rest. These findings likewise imply, that the time to incrementally bind a curve depends on the curve length. Therefore, decisions about the connectedness of two points will take longer the further apart these points are along the curve. This is in line with experimental findings on a behavioral level [12] as well as regarding the neural correlates of incremental binding [16].

### 2.2.2. Tracing operates on multiple scales—as predicted by the growth-cone hypothesis.
Consider once again two curves among which one needs to be selected and incrementally bound via attentional selection. As detailed above, the activity in the underlying neural contour representation is up-modulated to enhance the neurons' firing rate [11]. It was demonstrated [12] that the time to mentally trace a contour depends on the distance between the target and the distractor curve. The growth-cone model of object-based attention predicts that neural spreading can progress at larger scales when contours are positioned farther apart [14,17]. The hypothesis is that the faster spreading is accomplished by cells having larger receptive fields operating on a spatial grid that is sampled less densely by the neurons that perform the spreading. The spreading mechanism proceeds on the largest scale on which neurons represent the target contour unambiguously. There, ambiguity is determined by whether the contours captured by the same receptive field contain interfering distractors. For densely spaced contours, only cells with smaller receptive fields do not interfere and are able to resolve the spreading unambiguously [38]. The growth-cone behavior realizes a scale-invariant incremental binding process [14,17], a property of incremental binding that was as well behaviorally identified by earlier experiments [23].

To investigate this function in the proposed model, we more specifically performed experiments with respect to the growth-cone hypothesis of incremental binding. Thus, to vary the degree of target-distractor interference along the contour, we systematically varied the proportion of the overall stimulus which has a smaller distance between the target and distractor curve (Fig 5). In line with the idea of multi-scale incremental binding, the time until the whole curve is bound together depends on this narrow-wide proportion. The longer the narrow section of the configuration the more time it takes the model to reach the other end of the curve. This tracing time result already indicates that the

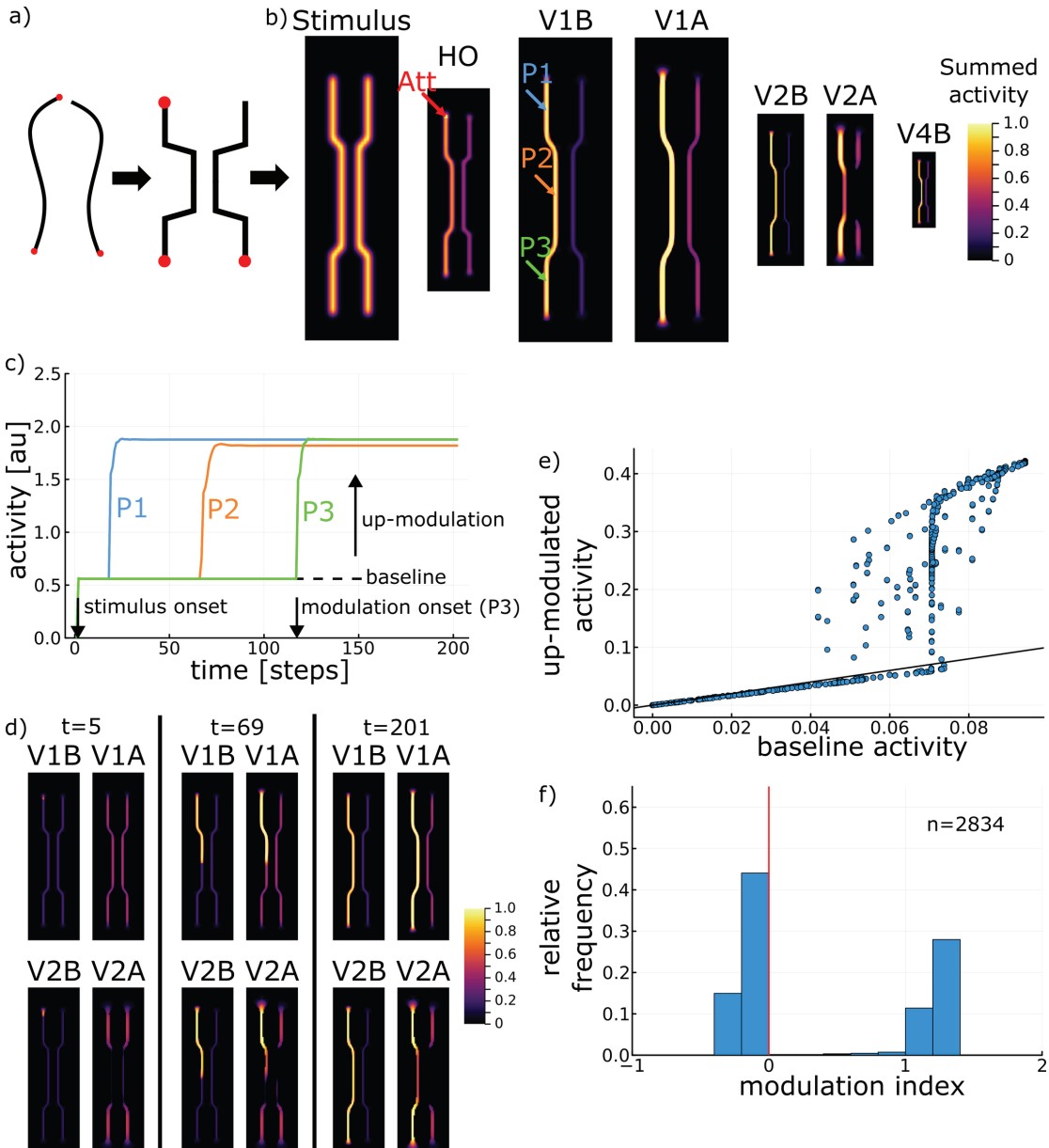

**Fig 4. Simulation results for stimulus with varying target-distractor distance.** (**A**) The synthesized stimulus captures the main properties of contour tracing stimuli [17] by providing a varying distance between target and distractor contour. (**B**) Neural representation after incremental binding has taken place for basal (V1B, V2B, V4B) and apical (V1A, V2A) potentials across model areas V1 to V4 and for higher-order thalamus (HO). Arrows denote the attentional seed location (Att) and probe locations (P1 to P3). Activity for areas V1 to V4 is summed across the feature channel and normalized to 1.0. Activity map sizes reflect the scale space ratios between areas. (**C**) Neural activity across time at the probe locations shows the main hallmarks of incremental binding, i.e., a constant up-modulation when being bound and a distance-dependent modulation-onset latency. (**D**) Neural representations at the beginning of (left), during (center), and after (right) incremental binding. While apical context of V1 is available for the complete contour from the beginning, the context of V2 is missing at first for the narrow contour segment due to interference. (**E**, **F**) Distribution of neural activities in V1 along the contour ($n = 2834$) while being grouped in comparison to their baseline activity (base grouping representation after stimulus onset, but before incremental grouping started), and distribution of modulation indices for the respective neurons. Neurons with a high baseline activity mostly experience a strong up-modulation during grouping, while a similar proportion of neurons ends up with nearly no modulation or slight suppression attributable to biased competition processes in the model (cf. Section 4.1.2). No modulation is denoted by the black identity line through origin in E and by the red vertical line in F. See Section 4.2.2 for details on the evaluation method.

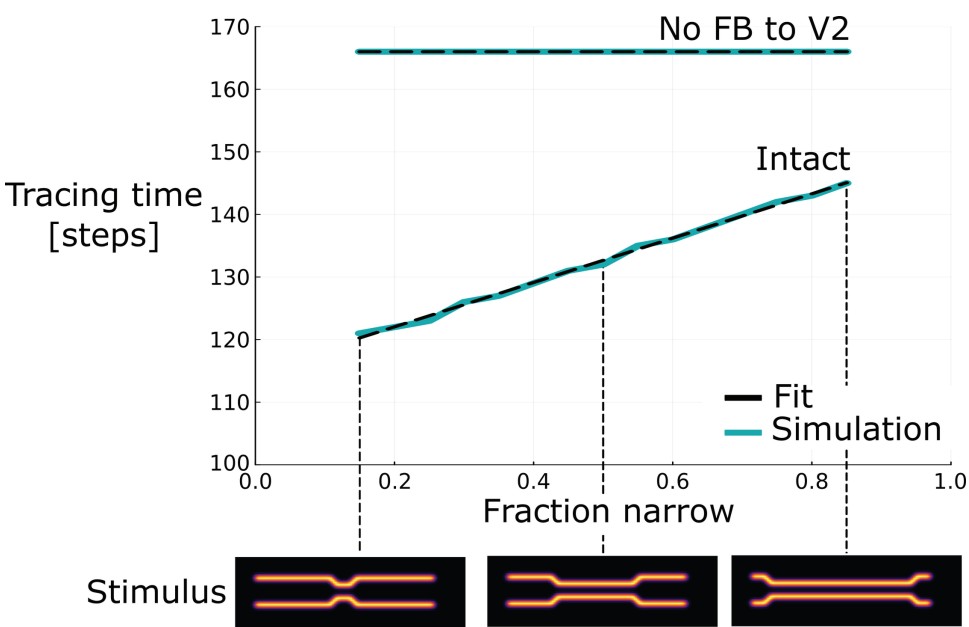

**Fig 5. Tracing time across input stimulus conditions of different distance proportions between target and distractor curves.** Tracing times for different proportions of narrow distances among target and distractor line segments for the intact model and a model without feedback connections from V4 to V2 (top). Representative stimuli for different such ratios are shown (bottom). The tracing time dependency on the fraction of narrow distances along the contour is well explained by a linear model (black, linear fit; blue, simulation data). In case of lesioning feedback connections of the higher hierarchical scales of the model no such distance-dependent tracing time variation is observed anymore. Instead, the tracing time becomes constant. Results extend a previous investigation that was performed with a simpler model version [46].

speed of the model's incremental binding computation depends on the target-distractor distance.

In addition to measuring the overall time to complete the binding process, the time it takes to label a segment of the contour can be computed. If the length of the respective contour segment is known in addition, then this yields an estimate of the binding speed for each segment separately. From these estimates a distance-dependent variability of binding speed can be seen from the modulation onset times of model neurons along the contour (Fig 6). There, a reduction of binding speed coincides with the narrow section of the curve configuration (2.14 neurons/step, 1.63 neurons/step, 2.12 neurons/step, for the first wide segment, the narrow segment and the second wide segment, respectively). This observation is likewise reflected by the apical integration process along the hierarchical scales (Fig 4D). While apical context information on the higher hierarchical scale can be established for the wide contour segments right from the beginning, this context information is only becoming available as the binding progresses for the narrow segments. Conversely, on the lower hierarchical scale this apical context information is established also for the narrow segments right away and readily can be integrated with the basal signal once a coinciding binding signal from the interfacing module become available. Thus, it can be seen in the model how this distance-dependent speed variation links to the apical representation of contextual feedback information.

**2.2.3. Ablation studies showcase model predictions.** The results so far established that the model is able to implement incremental binding and that these binding capabilities exhibit growth-cone-like variations in binding speed. There, the binding speed variations coincide

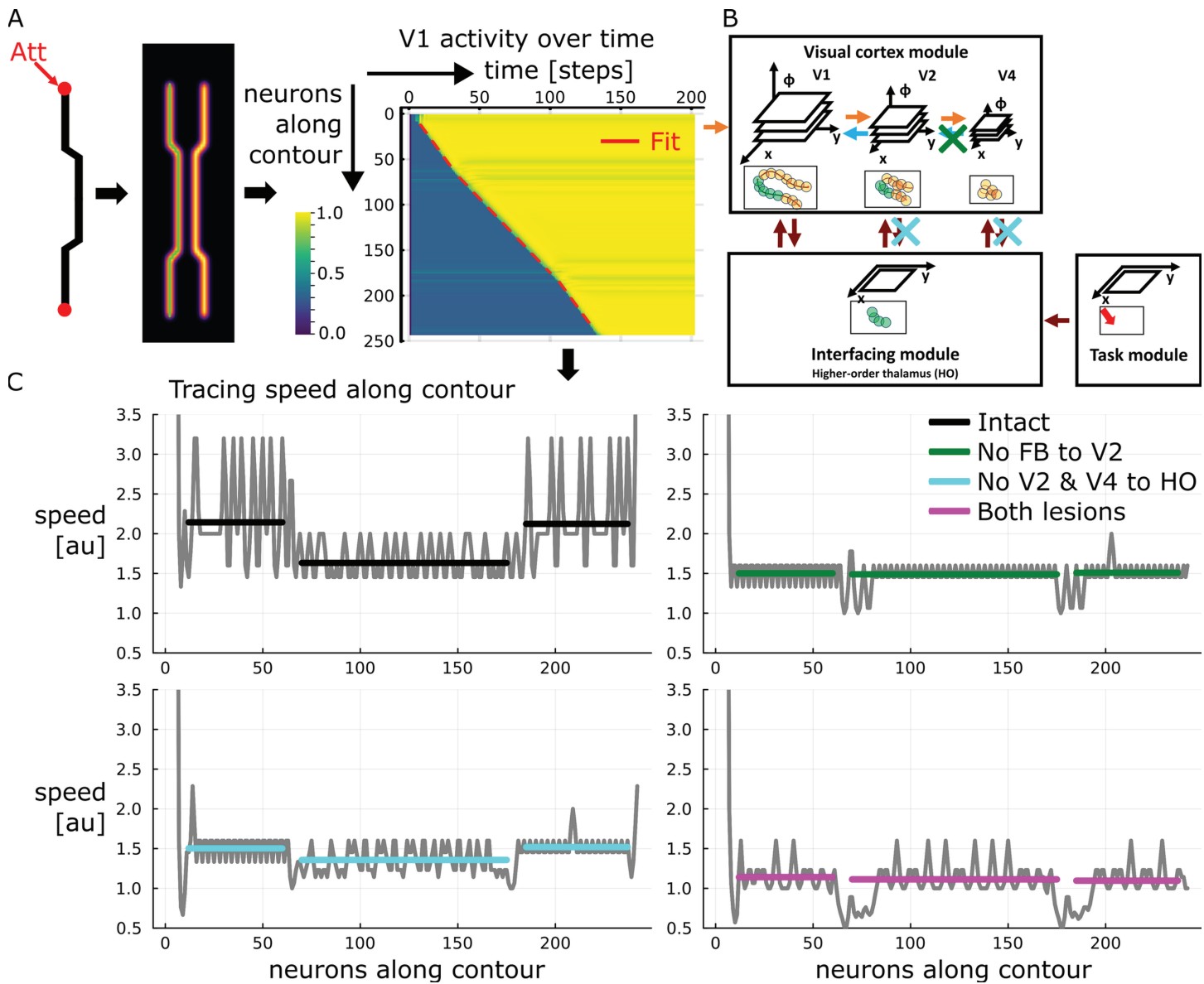

**Fig 6. Tracing speed varies along contour dependent on the lateral distance to distractor elements.** (**A**) Neurons along the contour show a modulation onset latency dependent on their distance from the attentional seed location. For the contour segments of constant distance to the distractor linear regressions provide an excellent fit to the onset latency along the line. The green line overlaid on the stimulus denotes the path along which neurons have been read out. V1 activity has been summed across feature channels and normalized to the maximum per neuron. (**B**) Based on the modular architecture specific connectivity patterns can be lesioned to investigate their functional role. (**C**) Computing smoothed finite differences between neighboring onset latencies provides an estimate of the instantaneous tracing speed at each neuron location along the contour. A higher speed indicates that a smaller time increment is necessary until a neuron becomes up-modulated if its neighbor has just become up-modulated itself. Transforming the regressed latency slope into an average speed per segment provides a good fit to the speed profile. Lesion studies yield speed profiles differing from the intact model version. The differences distinctly depend on the type of lesion. The different lesions are indicated in the architectural diagram. Gray lines denote speed estimates, black denotes the fit for the intact case, colored lines denote the respective fits for the different lesion cases. Deviations from the mean arise due to limited spatial and temporal resolution of the speed estimates (see Section 4.2.2 for details).

with a variation in the apical contextual state of neurons along the hierarchy of the visual cortex module. What cannot be seen directly from the results so far, is whether there is a causal link between the introduced scale space by the hierarchy and the binding speed variations, or whether the variation in binding speed arises within the model dynamics independently of the different scales. For that reason, we performed ablation studies to learn more about the functional relevance of the model's components. From such lesion studies it can be further deduced whether the model matches with experimental evidence and whether respective model predictions about impairments can be derived.

In order to selectively impair the complex information flow between the model's components, we eliminate connections between them. We investigated the effect of removing either specific feedback connections within the visual cortex module from area V4 to V2, or specific connections between the visual cortex module areas V4 and V2 and the interfacing module (Fig 6). These lesions specifically test for causal contributions of areas higher up the hierarchy of the visual cortex module to the model's incremental binding computation and by which pathways they impact the computation in which regard. Therefore, the lesions show whether these larger spatial scale representations contribute to growth-cone-like propagation speed variations in the model as would be predicted by the growth-cone hypothesis and has been observed in experimental data [14,17].

For investigating the impact of feedback connections, we removed the connections on the larger scale, i.e., from the highest area (V4) to the middle area (V2) in the model. Under this ablation condition the speed profile becomes near-constant along the curve's outline and, thus, misses the drop in speed for the contour segment with a narrow target-distractor distance (1.50 neurons/step, 1.49 neurons/step, 1.51 neurons/step). Furthermore, the constant speed level in this ablation case resembles rather the speed of the narrow than the wide segment of the contour's outline from the intact case (1.63 neurons/step vs. about 2.13 neurons/step, cf. above). Likewise, this constancy in binding speed can also be seen at the level of the total binding times, which do not vary with the proportion of narrow segment length (Fig 5).

This lesion case indicates that the model's feedback is necessary to exhibit target-distractor dependent speed-variations. Thus, the model's growth-cone behavior cannot be reproduced without feedback interaction at the coarser scales. Also, it can be seen that the target-distractor interference for the narrow segment during the intact case has nearly the same effect as removing the higher-level feedback during the lesion case, since both of these causes result a similar speed. That the interference for narrow segments leads to feedback that is virtually absent at the higher levels can also be seen again from the initial apical representation of the intact case (Fig 4D). This observation further demonstrates that the model implements growth-cone behavior by feedback among spatial scales and the interference thereof in case of close-by distractors.

For investigating the role of connections between the interfacing module and the higher areas of the visual cortex module we removed connections from both the middle area (V2) and upper area (V4) to the interfacing module. While an overall drop in binding speed can be observed, the qualitative speed profile stays to a degree intact exhibiting higher speeds for the wider segments of the contour's outline and a further reduced speed for the narrow segment (1.50 neurons/step, 1.36 neurons/step, 1.52 neurons/step).

The interfacing module accumulates evidence about up-modulated activity from all the areas of the visual cortex module and gates the subsequent binding increments. Thus, removing some of the inputs to this evidence accumulation, i.e., inputs from V2 and V4, slows down the process. As the inputs of larger scales affect a larger neighborhood than inputs from V1, not only the overall process is slowed down, but also part of the relative difference in speed

between wide and narrow segments is lost. So, in this case, the only scale at which evidence can be accumulated is the scale of V1. Yet, once enough evidence is accumulated from V1, a speed-variation is still noticeable, since the contextual integration of feedback from V4 to V2 is still intact. Once the interfacing module opens the gates and sends a task-relevance signal to the visual cortex module it allows also this higher-level (V4 to V2) integration to take place, which, in turn, affects the lower-level (V2 to V1) integration.

Applying both types of lesions simultaneously, i.e., removing feedback connections from V4 to V2 and connections from V4 and V2 to the interfacing module, results in a constant speed profile again. Notably, this constant speed is lower than any of the speeds for the individual lesions investigated before (1.14 neurons/step, 1.11 neurons/step, 1.10 neurons/step).

This lesion results in effectively decoupling the higher levels, and thus, scales, from taking part in the incremental binding process: higher-level contextual integration cannot occur due to missing feedback from V4 to V2, and neural activity in V2 and V4 cannot affect the evidence accumulation in the interfacing module due to missing connections. Thus, only V1 will be able to integrate contextual evidence from V2 and enter it into the interfacing module. Consequently, this lesion realizes the case of incremental binding on a single scale.

Together, we interpret these results as follows. The lesions indicate that both kinds of connections, namely cortico-cortical feedback connections and cortico-thalamic connections, influence the dynamics and speed of binding. Yet, specifically the feedback connections from model area V4 to V2 are responsible for the multi-scale effects associated with growth-cone-like incremental binding properties. Conversely, projections to the interfacing module from both model areas V2 and V4 affect the integration process overall and lead to a reduction in overall speed, if the projections are missing. In sum, a causal relationship exists between the model's feedback connections in scale space and its growth-cone-like binding properties.

**2.2.4. Task-dependent Gestalt disambiguation is based on attentional seeding.**  So far, the model has been tested on configurations with contours adjacent to each other. These configurations already revealed the model's incremental binding properties and its growth-cone-like behavior. There, the target contour was selected by the seed of attention on one of its ends which served as the fixation point. The influence of the other, distractor contour on the target contour's incremental binding process was mediated by the distance between the contours. What remains open from these investigations, is what happens to the model's binding process when target and distractor object intersect. Therefore, we chose a configuration with two contours crossing each other, similar to the stimuli used in [16].

We probe the model presenting pairs of contours which cross each other at about half the length of each contour (Fig 7). The attentional label again grows along the target contour until it reaches the location of the crossing point. Geometrically, the attentional label has three options to spread along one of the continuing arms that leave the junction or even tag multiple contour segments by splitting the attentional signal. The simulation demonstrates that the attentional label spreads along the smooth continuation of the contour at the entrance of the crossing. In other words, the incremental binding follows the Gestalt law of good continuation [4]. Beyond explaining the shown spreading behavior across intersections the model further generalizes to a range of different intersection angles between two curves (cf. S2 Text). We emphasize that incremental grouping in the model recruits mechanisms that are also used for linking perceptual items in base grouping operations.

By investigating the model's internal representation one can learn about why the binding happens like this (Figs 7 and 8). After the initial base representation has been formed by the

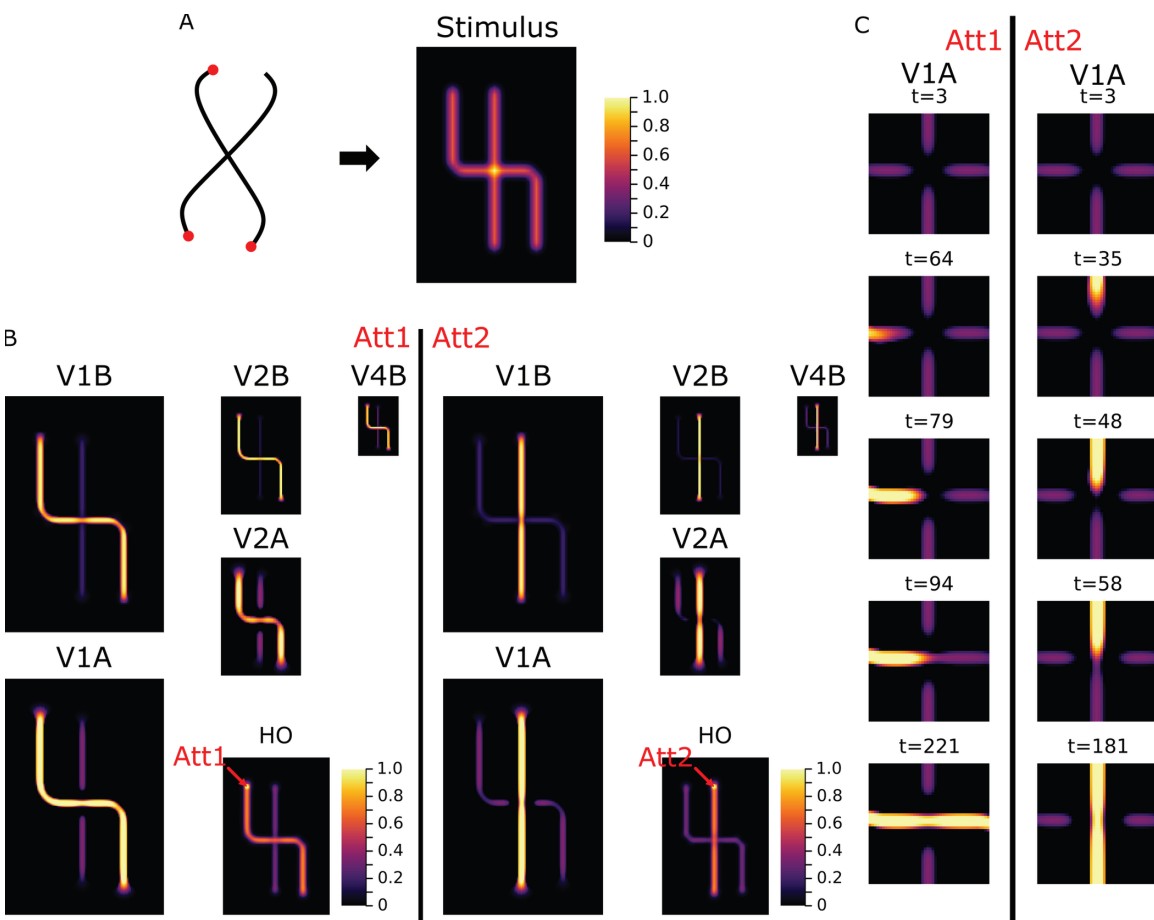

**Fig 7. Task-dependent grouping of an intersection.** (**A**) The synthesized stimulus captures the main properties of intersection stimuli [16]. The fixation point location can be varied between the two contours and steers the seed of attention to indicate which of the contours should be grouped into an object (Att1, Att2). (**B**) Dependent on the attentional seed location the simulation results in either one or the other line being grouped. The behavior follows the Gestalt law of good continuation. (**C**) Over the time course of integration, the role of the apical representation becomes visible. After the base representation has been formed the intersection is initially not represented apically in either condition (Att1 and Att2). This indicates no compatibility to either contour segments, because of mutual inhibition between the feedback projection to orthogonally oriented feature maps. Only once the up-modulation signal reaches the vicinity of the intersection, the mutual inhibition becomes biased and allows the intersection elements to become compatible to the respectively grouped line elements. Naming conventions and scaling of response maps according to Fig 4. See Fig B in S2 Text for further intersection stimuli.

model's feedforward processing, the aggregated apical contextual evidence around the crossing point exhibits a configuration with no preference for compatibility to any of the curve segments. Within the representation of the visual cortex module, neurons project to neurons in the next lower level of the hierarchy. There, they target neurons that are selective to similarly oriented contrasts via excitatory connections, and neurons that are selective to orthogonally oriented contrasts via inhibitory connections (Fig 8A and 8B; Fig A in S1 Text). The evidence for the four segments surrounding the crossing is equally strong encoded in the base representation across the maps of neurons tuned towards different orientations (Fig 8A and 8C). Thus, neurons at the location of the crossing, which are selective to either horizontally or vertically oriented contrasts, receive the same amount of excitatory and inhibitory feedback projections from the likewise and orthogonally tuned neurons of the higher level, respectively. As

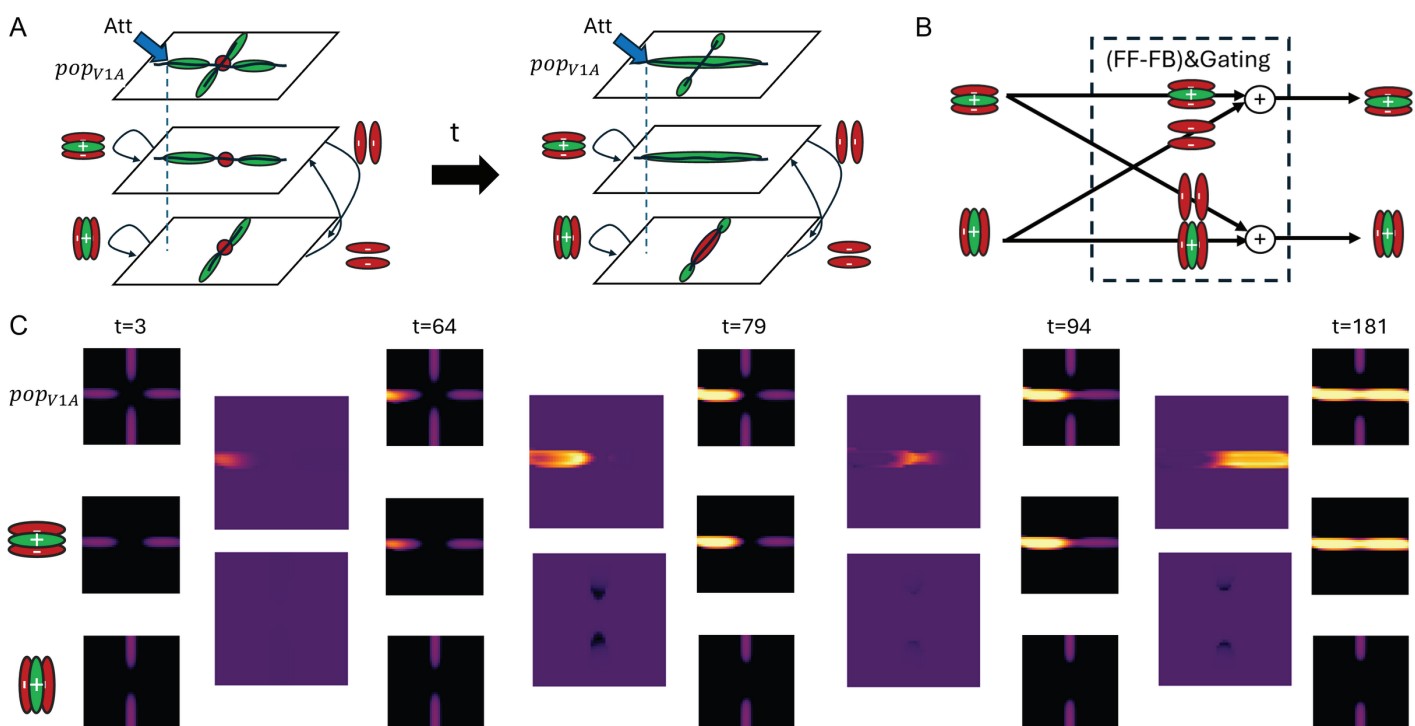

**Fig 8. Task-specific apical integration for intersecting lines.** (**A**) Schematic depiction of the effective reciprocal influences between different maps of orientation-selective cells of a common population of area V1 apical compartments for the case of two intersecting lines. We show two phases as samples of the continual integration process. During an initially ambiguous phase horizontal and vertical influences balance out (left). During a later phase a disambiguated state emerges through the bias generated by attentional signals (right). Based on feedforward-feedback filtering between neurons in V1 and V2, neurons in different orientation maps provide enhancing (similar tuning and coherent position; green) or suppressing (dissimilar tuning and incoherent position; red) influences on the development of apical contextual potentials. Biasing attentional signals unspecific to orientation selectivity (blue arrow) work on these mutual interactions to establish a new, task-specific equilibrium. (**B**) Diagram that depicts the influences different orientation-selective cells exert on the development of apical potentials of cells with same and orthogonal orientation tuning. Forward-feedback filtering between adjacent hierarchical areas is realized depending on the specific combination of orientations. This establishes an interaction skeleton between features to express current compatibilities ([15]; cf. Fig A in S1 Text for feedback filters). Gating input from the interfacing module allows or prohibits the contextual influence on cell output independent of tuning, but dependent on the attentional state. (**C**) Temporal evolution of apical potentials for the case of an intersection with attention directed towards the horizontal line. Apical potential is aggregated across orientation maps (top) and is separately visualized for horizontal and vertical channels (mid, bottom). Larger displays between timesteps of evolution depict differences in apical potentials between adjacent panels. An incremental progression of apical amplification from left (initial seed of attention) to right (final system state) can be noted for the horizontal channel, a suppression of apical potentials close to the crossing's center is visible for the vertical channel. In combination, both influences establish a disambiguated version of an apical interaction skeleton, which allows the attentional signal to continue spreading horizontally beyond the crossing.

a consequence, the initial apical evidence at the crossing location is balanced for horizontal and vertical preference. These equal contributions only change later on, once the incremental binding signal spreads into the vicinity of the crossing point's representation. Then, neurons coding for the orientation of already bound segments are up-modulated. Their enhanced activity is subsequently fed back onto their lower-level neighbors and biases their apical input towards oriented contrasts collinear to those of the already bound curve segments, similar to association fields ([63,64]; cf. S1 Text for further details on apical coupling structure). Through this interaction, the model establishes an incremental binding version conforming not only with neural correlates of incremental binding and the growth-cone hypothesis, but also with the Gestalt law of good continuation [4] in the crossing contours case.

**2.2.5. Model performance on more complex stimulus geometries.** Up to this point, we investigated the function of the model using line stimuli with varying distances and with and without crossings (as in experiments, [16,17]). We conducted additional simulations on

a set of other, more complex, configurations inspired by further psychophysical experiments (Fig 9, see S2 Text for supporting information). These simulations show whether the model generalizes to other stimuli and also reveal its limitations.

The stimuli test the model performance regarding different spatial input configurations and item features. We specifically investigate phenomena of similar tracing speed on

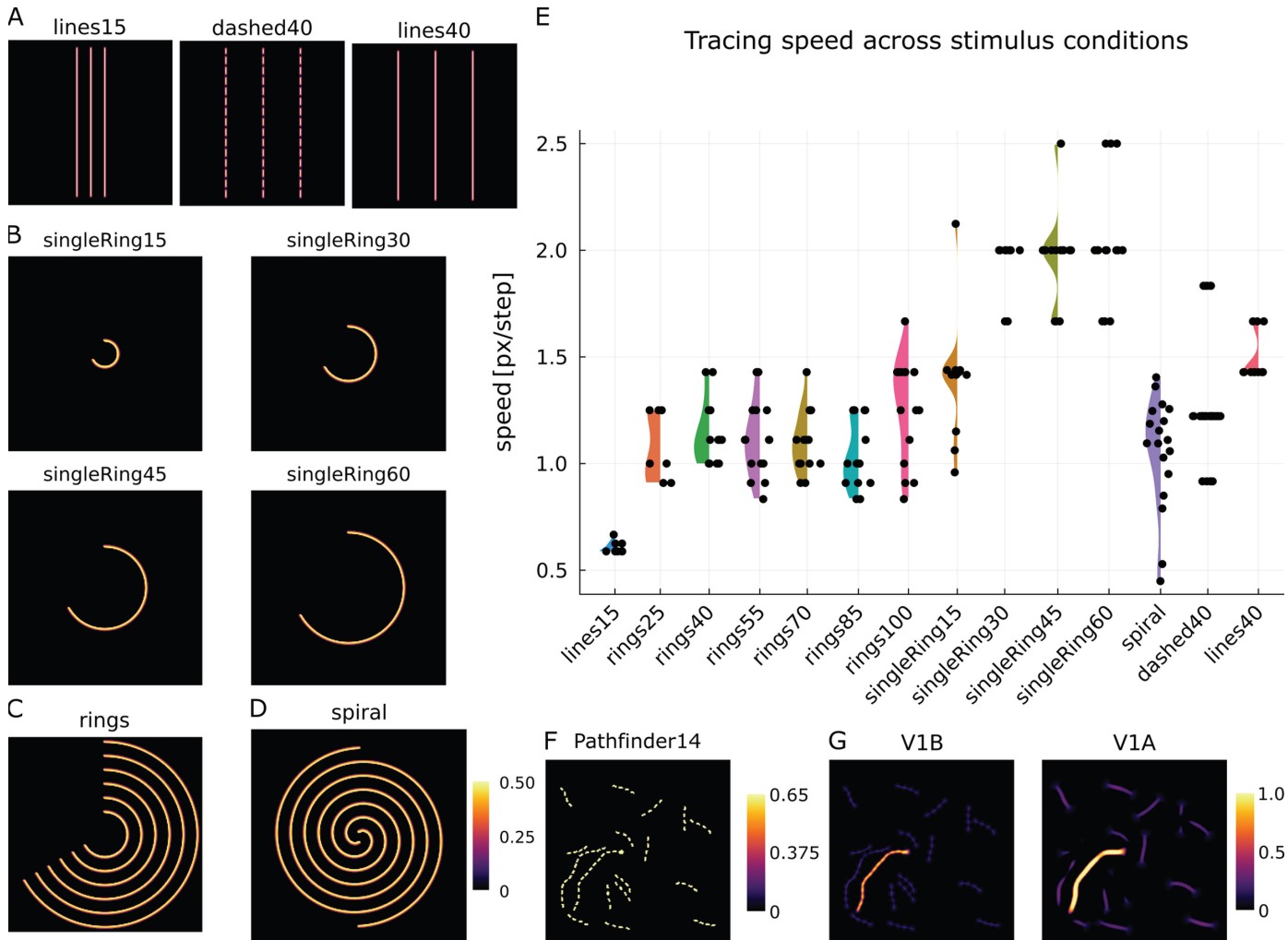

**Fig 9. Model performance for a broader set of stimuli.** The model successfully performs tracing on different types of stimuli and shapes. (**A**) Line configurations with a small (15 pixels) distance between lines (`line15`) and a larger distance (40 pixels) between lines for the case of dashed line segments (`dashed40`) and solid lines (`lines40`). (**B**) Single ring segments of 240 degrees with different radii (15 pixels, `singleRing15`; 30 pixels, `singleRing30`; 45 pixels, `singleRing45`; 60 pixels, `singleRing60`). (**C**) Configuration consisting of 6 rings with radii between 25 and 100 pixels in steps of 15 pixels. Attentional seed location can be varied (e.g., `rings25` denotes attentional seeding of innermost ring). (**D**) Configuration of two interleaved spirals with a radial spacing of 15 pixels between neighboring lines and a starting radius of 6 pixels (`spiral` condition). (**E**) Distribution of tracing speeds for the different stimuli as measured by probe locations along the curves. Speeds are comparable among fixation locations in the ring condition and larger than in the lines condition with comparable distances between lines (`lines15`). If no other rings are acting as distractors (`singleRing` conditions) tracing speed is faster than if distractors are present (`rings` conditions). Only for very large curvatures (smallest radius tested) a decline in speed is noticeable (`singleRing15`). The `spiral` condition contains a range of tracing speeds but reaches a level comparable to the one of the rings conditions. While an influence on the exact distribution of speeds along the extent of the broken lines is noticeable (`dashed40`), the range of speeds is comparable to that of the solid lines (`lines40`) case. (**F**) Example stimulus from the `Pathfinder14` dataset [65,66]. This more realistic stimulus combines challenges of tracing on changing orientations, curvatures, and dashed contour elements. The target curve is indexed by a bright dot. (**G**) Model simulation on the `Pathfinder14` stimulus (V1 basal and apical potentials aggregated over orientation channels). Depicted stimuli are scaled in proportion with respect to each other. See S2 Text for supporting information.

broken versus solid lines [5] (`dashed40`, `lines40`, Fig 9A); reduction of tracing speed with increasing curvature and relation to straight lines with similar distractor spacing [35] (`singleRing`, `rings` and `lines15` conditions, Fig 9A, 9B, and 9C); ability to trace on more complex stimuli with intertwined distractors [23] (`spiral`, Fig 9D); and extensibility toward more realistic stimulus configurations (`Pathfinder14`, Fig 9F).

Comparing the tracing speed on broken versus solid lines, one can see that the range of speed values is similar for both cases (Fig 9A and 9E; `dashed40` versus `lines40`), while being slightly larger on average for solid lines (cf. Table A in S2 Text). On the set of rings, the dependency of the model's tracing speed on curvature is investigated (Fig 9B and 9C). The presentation of individual rings of different curvature therefore only impacts tracing speed for the most extreme curvature tested (`singleRing15` stimulus), but yields a similar tracing speed for smaller curvatures, i.e., larger radii (Fig 9E; `singleRing40-singleRing60`). Likewise, having nearby distracting rings affects the tracing speed, but does so seemingly independent of the actual curvature of the traced ring (Fig 9E; `rings` cases). Notably, the tracing proceeds also faster than for a configuration without curvature but similar line distances (`lines15` stimulus; Fig 9A and 9E). That the model is able to trace on curved outlines with complex involvement of distractors can also be seen on a spiraling curve, where radius, and, thus, curvature, continuously varies along the curves (Fig 9D and 9F; `spiral`). There, close to the spiral's center the model first traces with a low speed and becomes increasingly faster until it reaches a similar distribution of speeds as the `rings` stimulus (Fig 9E; cf. S2 Text for a tracing speed profile). A stimulus from the Pathfinder dataset shows a combination of the former stimulus features ([65,66]; `Pathfinder14`, Fig 9F). The dataset consists of randomly generated dashed lines with different lengths and changing orientations. Among these, one line is the target line indexed by a bright dot. The dataset was designed as a test bed for learning long-range interactions in deep neural network models, that at the same time serves as a precursor for model deployment to realistic scenes. The proposed model successfully traces the target contours on a proportion of the dataset (Fig 9G). The success of the tracing on stimuli from the dataset depends on the respective target-distractor configuration. If local curvature of the target line is too high, or the distance to co-linear distracting lines is too small tracing will stop and fails to cover the complete target line.

With this set of experiments the proposed model is evaluated for how well it can explain a broad range of experimental evidence. Overall, the model is able replicate many aspects from these findings. It exhibits a similar tracing speed on dashed versus solid lines ([5]; `dahsed40`, `lines40`) and shows to some extent the reduction in tracing speed under strong curvatures ([35]; `singleRing`). The model is likewise able to perform tracing on more complex stimuli ([23]; `spiral`) indicating that it generalizes beyond simple straight-line configurations. This generalization capability is in part shown by the model's tracing performance on a subset of stimuli from the Pathfinder dataset ([65,66]; `Pathfinder 14`). Yet, also limitations of the proposed model can be seen on the Pathfinder dataset, as tracing success depends on the respective target-distractor configuration. Furthermore, experimental findings would predict a curvature-dependent reduction of tracing speed for curved lines under the presence of distractors ([35]; `rings`). There, the model so far only shows constant tracing speeds.

## 2.3. Complexity considerations of incremental binding architectures

The proposed model links on an implementational level descriptions from pyramidal neurons to inter-areal computation and provides a dynamical realization of the incremental binding algorithm. Beyond the simulation results that show that the model is able to explain a variety

of experimental evidence, a theoretical analysis of the model's complexity yields insight into how far it fulfills a normative account. Such theoretical complexity analyses have been previously constructed to constrain theories about computational processes underlying visual function [19,67,68]. To this end, we analyze the model's complexity in terms of connectivity and representation and compare it against alternative realization schemes for solving incremental binding. This analysis extends the canonical framework of incremental grouping by Roelfsema [15] and builds upon earlier investigations of our own [46,47].

### 2.3.1. Basic components for incremental binding.

The analysis focuses on incremental binding models that capture the neurophysiological evidence discussed above. Thus, models need to be able to compute a base representation and an incremental representation, and they need to do so within a hierarchy of multiple scales to achieve growth-cone-like incremental binding. Abstractly, such models can be described by network graphs [15]. These network graphs are composed of two basic elements: neural processors (NProcs) and channels. *NProcs* capture the computation that takes place within a local microcircuit at a certain retinotopic position and hierarchical scale. *Channels* capture the network's connectivity structure and form passive elements that relay information between NProcs unidirectionally. Different models of incremental binding can be described by these elements. Such models can be distinguished by the properties of the network graph, i.e., by how NProcs encode the representational information, what types of channels exist, and how they connect NProcs with each other.

Such distinction between models helps categorizing them with respect to their computational properties and complexity. A comparison based on the complexity properties additionally answers the question to what extent a model not only provides a mechanistic account for incremental binding, but also a normative one.

The following subsections investigate the complexity of individual network components in more detail. First, different variants of how NProcs can encode the necessary information are discussed with respect to their representational complexity. Second, it is investigated how different channel types can be used to provide NProcs with the necessary information for their computation and what it means in terms of connectivity complexity. Third, it is shown which possibilities exist to interface with these networks in terms of task signaling and read-out and what are their pros and cons. The last subsection then relates these insights to the specific model proposed in this work.

### 2.3.2. Local neural processors and their representational complexity.

To perform incremental binding, models need to adhere to the main phases of the incremental binding algorithm, i.e., computing a hierarchical base representation at first and an incremental representation afterwards. Each NProc is involved in this computation locally and needs to implement decision rules accordingly. A symbolic interpretation of these decision rules provides an understanding of the required flow of signals within the network. On a local NProc level this computational logic can best be understood in terms of Boolean logic using "and" ($\wedge$) operations. Similarly, a population level view across NProcs can be used to describe the incremental binding algorithm. There, set-theoretic terms describe the algorithm, where becoming a part of the set of NProcs coding for a specific variable would translate into a union ($\cup$) operation. In this contribution the necessary computational logic will be described in local Boolean terms to understand the decision functions each NProc needs to implement locally.

For computing the hierarchical base representation NProcs need to have access to the base evidence provided by the respective lower hierarchical level (or by the visual input directly in case of the lowest level's NProcs). If sufficient evidence is provided to the respective NProc it should likewise code for this evidence at its output. In Boolean terms, each NProc that is

part of the base representation computes a binary variable about the evidence for base representation at its position $X_{Base} \in \{noevidence, evidence\}$ as a function of the incoming base representation variables $X^*_{Base}$ from other NProcs $X_{Base} = f(X^*_{Base})$.

To compute whether an NProc belongs to the incremental representation or not it needs to have access to two distinct signals: the base representation for the space-feature combination for which it is selective at the respective hierarchical scale, and the incremental representation from its neighboring NProcs. If enough evidence for the local NProc's base representation is available and the incremental representation among its neighbors provides a compatible constellation for this NProc, then it should likewise become a part of the incremental representation and code for this circumstance at its output. In Boolean terms each NProc that is part of the incremental representation needs to compute a binary variable about its binding state, i.e., $X_{Binding} = \{unbound, bound\}$. This variable is computed by a logical "and" between the NProc's base evidence $X_{Base}$ and the evidence about the incremental representation, which is a function of the binding state variables $X^*_{Binding}$ of the neighboring NProcs $X_{Binding} = X_{Base} \land f(X^*_{Binding})$.

Different alternatives exist to implement this Boolean NProc logic: a naive *two-node static* implementation and a *single-node static* implementation which employ a static encoding, and a *single-node phasic* implementation which relies on bi-phasic codes.

A naive *two-node static* implementation of these decision rules above will require $log_2 4\,bits = 2\,bits$ to encode the two binary variables $X_{Base}$ and $X_{Binding}$ as independent output signals. In this case a model either assumes to have two NProcs per space-feature combination with a binary output code for $2 \times 2$ distinct values, e.g., one coding for the base representation and one coding for the incremental representation, or to have one NProc that provides a more complex output code with 4 distinct values to cover all combinations between the variables. An improved *single-node static* version considers the fact that the binding state $X_{Binding}$ is dependent on the base representation $X_{Base}$, in such a way that it can never be in binding state if no base evidence exists for that space-feature combination, i.e., $X_{Binding} = bound$ if $X_{Base} = noevidence$ is an impossible state. Thus, effectively only 3 distinct values $\{noevidence, evidence \land unbound, evidence \land bound\}$ need to be encoded at the output. As the binding state implies the existence of base representation evidence the value can simply become $\{noevidence, evidence, bound\}$ (Fig 10A). This leads then to $log_2 3\,bits \approx 1.58\,bits$, which could be encoded by a single NProc with a ternary output code. Another alternative exists in the *single-node phasic* implementation if one assumes a strict phasic processing of the two representations. Then, instead of having two binary NProcs for $2 \times 2$ distinct values, one such binary NProc would suffice coding during the first phase for the base evidence and then during the second phase for its binding state. Yet, to do so it would require additionally a memory of 1 *bit* to store the information about its base evidence state from the first phase.

As a result, the ternary NProc realization is the most efficient one in terms of bits to encode the required representations.

**2.3.3. Connectivity schemes and their complexity.** To compute their outputs NProcs need access to the signals from other NProcs for base evidence $X^*_{Base}$ and binding state $X^*_{Binding}$. The transmission of these signals between NProcs happens via unidirectional channels. The necessary connectivity scheme for these channels depends on the signal they transmit. For base evidence signals it is sufficient to connect from NProcs of adjacent scales of the hierarchy (from lower to higher; Fig 10B). As a result, each NProc needs to have channel connections with NProcs from $K_{l-1}$ neighboring locations from the lower hierarchical scale $l-1$ and $N_{F,l-1}$ features per location. For each location, this connection scheme must be established

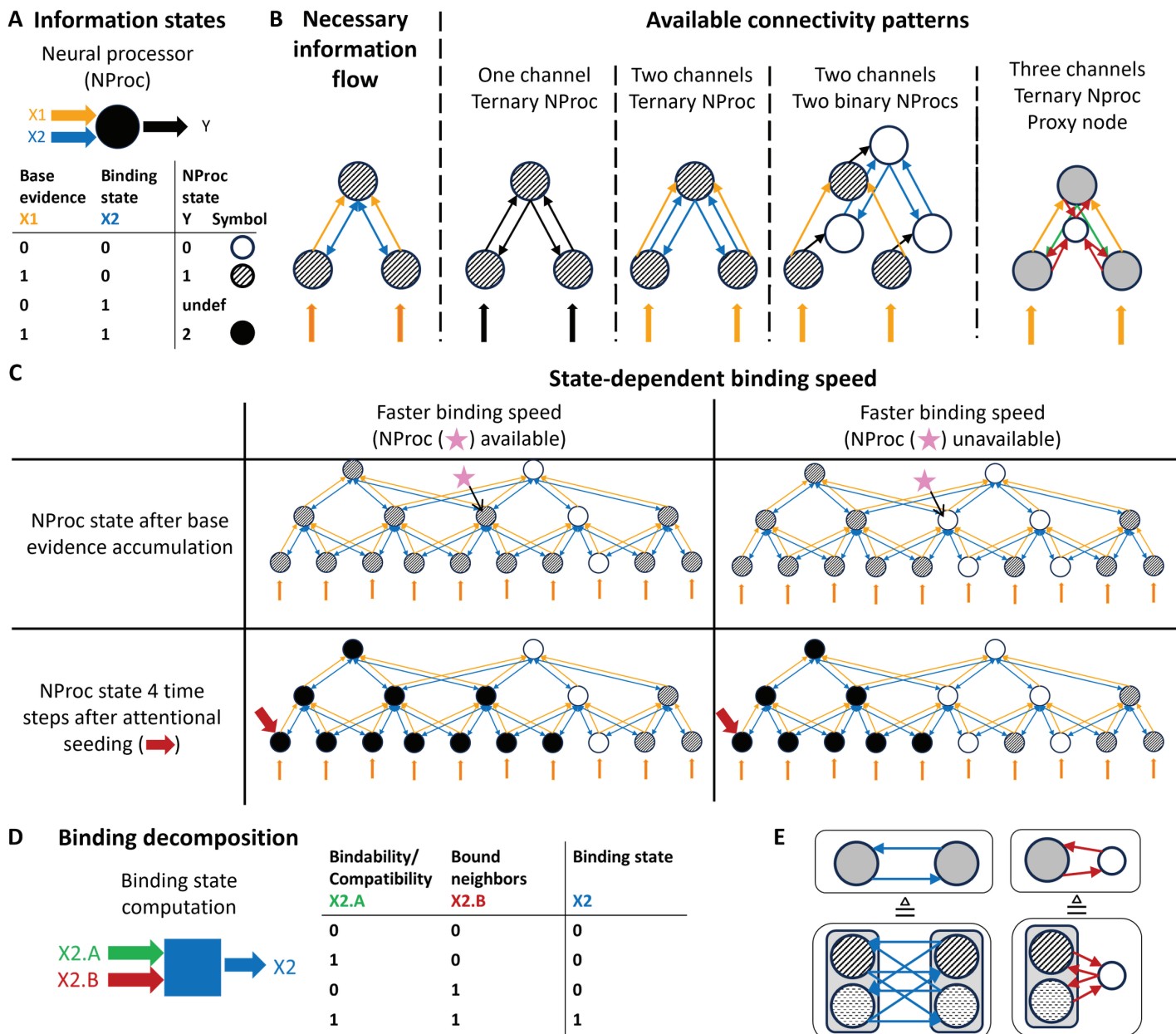

**Fig 10. Incremental binding framework and complexity considerations.** Binding computation can abstractly be represented by a graph with nodes coding for information and edges propagating specific semantics. (**A**) Each neural processor (NProc) captures the processing of a local neural processing site. NProcs encode semantic information states: no evidence (white filling), existing base evidence state (textured filling), or a binding state (black filling). (**B**) Necessary *bi-directional* information flow among NProcs to realize incremental binding (left) and available connectivity patterns to implement this information flow using *uni-directional* couplings (right). The proposed target architecture uses the "Three channels, ternary NProc, Proxy node" pattern (last column). (**C**) State-dependent binding speed based on a hierarchical incremental binding architecture as proposed by Roelfsema & Houtkamp [15]. The graphs conceptualize the cases of faster (left column) and slower (right column) binding speeds, respectively. Here, the connectivity pattern for ternary NProcs is shown. After an initial phase of evidence accumulation (first row) incremental binding can take place among the hierarchy starting from an NProc for which an attentional seed (red arrow) has been provided. Combining base evidence and neighborhood information, specific NProcs among the hierarchy become (un-)available (indicated by pink star) and speed up or slow down the binding process. (**D**) Two decomposed sub-signals can be joined by an "and" operation to retrieve the binding state information of an NProc. (**E**) Main connectivity variants underlying the complexity considerations. If NProcs are extended for feature selectivity per location (denoted by different fillings), full connectivity is required to compute the binding state information (left). Using a proxy node to aggregate the binding state information across features greatly reduces the number of connections (right; see text for details).

for all NProcs coding for $N_{F,l}$ different features. Thus, the required amount of base evidence channels per location is characterized by $N_{BE,l} \sim O_{BaseEvidence}(K_{l-1} \cdot N_{F,l-1} \cdot N_{F,l})$.

For binding state signals the situation is more complex. On the one hand, NProcs need to compute the binding state given the context of their neighbors. This context can be established by channels laterally connecting from neighbors within a hierarchical level, or by feedback connecting from neighbors of the next higher hierarchical level. On the other hand, incremental binding proceeds as a growth-cone-like process. This requires that NProcs need to communicate their binding state signals back and forth across the hierarchy to other NProcs that are selective for the same space-feature combination but on another spatial scale (Fig 10C):

- If spreading has been taking place faster at a higher hierarchical scale, the corresponding NProcs at the lower scales need to receive feedback to enter the binding state signal as well on their level of representation.
- If spreading has been taking place on a lower hierarchical scale but the local context of an NProc on a higher hierarchical scale would now allow for the binding state, then the lower level's binding state signal needs to be transmitted to the higher level via feedforward connections.
- If spreading can only continue on the current scale, then lateral connections would be required to signal this information across neighbors and allow them to integrate this evidence with their contextual information.

Thus, the naive realization of binding state channels would require all three kinds of connectivity schemes, feedforward, lateral, and feedback. The required amount of binding state channels per location in hierarchical level $l$ is then characterized by $N_{BS,l} \sim O_{BindingState}(N_{F,l} \cdot P_l)$, where $P_l = K_{l-1} \cdot N_{F,l-1} + K_l \cdot N_{F,l} + K_{l+1} \cdot N_{F,l+1}$ denotes the pool of $K_l$ neighboring NProcs from the different levels $l$ of the hierarchy.

Here, a simplification is possible. It approximates the connectivity scheme by assuming the binding state signal is (de-)composable. Instead of having to compute the binding state signal directly from its neighbors ($X_{Binding} = X_{Base} \wedge f(X^*_{Binding})$), an NProc might compute two intermediate signals for whether it possesses a bound neighbor $X_B = f_{Bound}(X^*_{Binding})$ and whether its own space-feature combination is compatible to its local neighborhood $X_C = f_{Compatible}(X^*_{Binding})$. If both signals are available, then the binding state signal can be composed. In Boolean logic this would correspond to an "and" function once again $X_{Binding} = X_{Base} \wedge f(X^*_{Binding}) = X_{Base} \wedge (X_B \wedge X_C)$ (Fig 10D). This simplification assumes independence of the binding state signal and the compatibility signal, i.e., the neighbor which provides the information that a binding signal is available does not need to belong to the same neighborhood from which the compatibility of an NProc to the bound representation is calculated.

To extract the information whether a bound representation exists in the neighborhood, the binding state information can be provided by any neighbor irrespective of the feature selectivity. Since it is this binding information which needs to be propagated across all hierarchical scales to establish a growth-cone-like algorithm the decomposition approach can save connections of all three kinds across the hierarchy, i.e., feedforward, lateral, and feedback. What is required, though, is the introduction of an intermediate proxy node, which aggregates the binding state information across features of a local neighborhood to obtain the featureless binding state signal (Fig 10B and 10E). Then, the connectivity complexity can be reduced greatly to $N_{B,l} \sim O_{BindingState}(N_{F,l} + P_l)$ from $N_{F,l} \cdot P_l$ (cf. above). Conversely, whether

the neighboring bound representation is compatible with an NProc still requires an evaluation of feature-based compatibility. Yet, by the simplification, it suffices to compute the compatibility information from one of the neighborhoods, i.e., from the next lower, same, or next higher scale, respectively. Thus, the complexity of evaluating the compatibility still needs to be based on evaluating an NProc's feature against all other features of the neighboring locations, but now the neighborhood information only needs to stem from one neighborhood source, e.g. via feedback channels, leading to $N_{C,l} \sim O_{Context}(N_{F,l} \cdot (K_{l+1} \cdot N_{F,l+1}))$ with $K_{l+1} \cdot N_{F,l+1} < P_l$.

This complexity reduction by decomposition yields the greater benefits in terms of connectivity the more neighbors $\{K_{l-1}, K_l, K_{l+1}\}$ each NProc possesses and the higher-dimensional the feature space $\{N_{F,l-1}, N_{F,l}, N_{F,l+1}\}$ is. Thus, these benefits are one of the major motivations for the model architecture proposed here.

**2.3.4. Interfacing demands for incremental binding as an elemental operation.**  An additional requirement for models of incremental binding is imposed by the fact that incremental binding, as an elemental operation, needs to be employed within more complex visual routines and cognitive programs. Such programs are supposed to achieve a behavioral goal, e.g., making a saccadic eye movement to signify a decision about the task. This requires the incremental binding implementation to interface with other operations and to be composable into visual routines. Consequently, the question becomes relevant about how the incremental binding operation can be interfaced efficiently. In other words, this question addresses the signature of the operation with its input and output interfaces. Regarding the input, the operation can be triggered by providing it with an attentional seed location. Regarding the output, the operation's result can be read out in terms of the bound representation.

Under the growth-cone hypothesis, the relevant NProcs to interact with, as targets for input and sources for output, can be found replicated across the different hierarchical scales of the model. Thus, different options exist to realize such interface (Fig 11). Besides the necessity to function under all circumstances, such option should be favorable that requires the least number of connections. The extreme cases of realization are top-level interface, bottom-level interface, and full interface. The *top-level interface* uses the representation at the highest hierarchical scale of the model as an interface to provide the attentional seed and to perform a read-out of the binding state (Fig 11A). Since the highest scale offers the most compact spatial representation due to the spatial down-sampling, this realization would yield a low number of connections. The *bottom-level interface* uses the opposing end of the model's hierarchy as an interface sending information to and reading from the lowest hierarchical scale (Fig 11B). Since this is the most extensive spatial representation due to its high resolution, this realization would yield a high number of connections. The *full interface* realization uses all scales of the model hierarchy as representations to interface with (Fig 11C). As a result, this realization will yield the highest number of connections. Considering only the number of connections the top-level interface would provide the favorable solution. Yet, there is a downside: If models should implement the growth-cone hypothesis, then processing of the binding state information can only proceed up to that level of the hierarchy which provides an interference-free representation between the target object's components and distractors. Likewise, providing an attentional seed via the interface would fail in such cases for which interference would occur at that level to which the attentional seed has been provided. An analogous argument holds for reading out the binding state from such scales. Thus, only a bottom-level or full interface are options for a robust realization of incremental binding under all circumstances.

An additional complexity reduction step that can be applied to either realization consists in the usage of a *proxy interface* (Fig 11). While a cognitive controller would have more degrees of freedom if it would interface with the NProcs of a respective scale directly, it would also

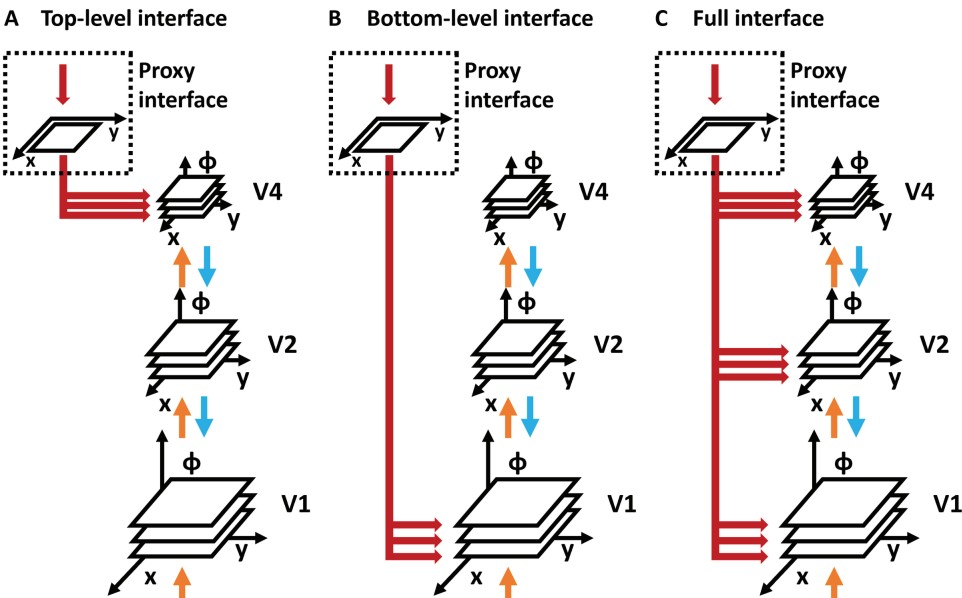

**Fig 11. Variants of interfacing with the incremental binding network architecture to provide an attentional seed from a task module to different hierarchical levels.** (**A**) A top-level interface restricts provision of the seed of attention to the highest level of the hierarchy. (**B**) A bottom-level interface restricts provision of the seed attention to the lowest level of the hierarchy. Both interface variants form extreme cases of a range of interface patterns. (**C**) The most general, but also connection-wise most expensive variant is formed by the full interface variant. Irrespective of the interface variant the number of connections that directly need to originate from and be controlled by a task module depend on the number of features per retinotopic position in the hierarchy. A proxy interface can be utilized as a solution. There, only positional information about the attentional seed is provided by the task module to a proxy map of purely spatial encoding nodes and broadcast to all space-feature encoding NProcs of the associated hierarchical level. The proposed model architecture utilizes the proxy interface pattern in combination with the full interface variant. Additionally, it establishes a bi-directional interaction with the proxy map to realize an information flow pattern which reduces the connectivity complexity for incremental binding further (see text for details). Colors as in Fig 3.

need as many direct connections as there are NProcs per feature and location. Yet, if the cognitive control mechanism only needs to interact with spatial information, i.e., to which location the seed of attention needs to be allocated, and to read out which locations are occupied by the bound object, then the cognitive controller does not need such fine-grained control over features. Instead, a proxy interface can encode all spatial locations within a map without a replication across features. It can then provide an interface for the control mechanism and broadcast the attentional seed to all NProcs of a scale coding for the different features at the corresponding location. Likewise, such proxy interface can aggregate the binding state from all NProcs of a scale by marginalizing over the feature dimension and only retain the location of the binding states. While such proxy interface comes at the cost of adding another set of nodes for the spatial map representation, it reduces the representational complexity of the interface with which a cognitive control mechanism or other areas of the brain would have to interact.

Consequently, the proxy interface uses $N_{Inj} = L_p + M$ connections to connect the cognitive control mechanism with the levels of the hierarchy targeted by the respective injection method. There, $M = \sum L_l \cdot N_{F,l}$ is the number of connections needed to connect the interface to NProcs of each space-feature combination across all targeted levels $l$ along the hierarchy, i.e., $L_l$ is the number of positions and $N_{F,l}$ the number of features in a level. This

number $M$ is identical to the number of connections a respective injection method would require without the proxy interface. Conversely, the proxy method needs additional $L_p$ connections to connect the cognitive control mechanism with the interface, where $L_p$ is the number of spatial positions encoded by the interface. Thus, while the total number of connections and nodes increase by $L_p$, the number of connections which are required from the cognitive control mechanism itself reduce from $M$ to $L_p$. This reduction becomes greater the more levels $l$ are targeted by the injection method. While the greatest savings are established for the full injection case, already the case of targeting a single level results in a reduction from $L_l \cdot N_{F_l}$ to $L_p$.

 **2.3.5. Resulting model complexity.** The proposed model combines the optimal complexity cases for each of the components discussed above and further integrates them with each other. Thus, it establishes a normative account for the considerations above. It combines the different optimal components from the considerations of NProc coding, channel connectivity and interface realization. Thus, the presented model uses NProcs with a ternary code mapping the logical states, $\{noevidence, evidence, bound\}$, to the gradual output regimes of the pyramidal cell model of zero firing rate, low to intermediate firing rates and high firing rates, respectively. To stabilize these gradual output regimes, pyramidal cell model neurons compete via inhibitory pooling across a local space-feature neighborhood. The pyramidal cell model utilizes feedforward channels for base evidence signals entering the pyramidal cell's basal compartment and feedback channels for the compatibility computation entering the cell's apical compartment (cf. Fig 3). The binding state information is stemming from proxy nodes marginalizing binding state information across features per location. It enters the pyramidal cell computation via a gating component effectively realizing the required "and"-logic. The interface is likewise realized by making use of such proxy nodes. Yet, instead of utilizing a different set of nodes for the interface, a further simplification is performed by using the same proxy nodes for interfacing and model-internal binding state transmission. Therefore, the required number of proxy nodes is further reduced by $\sum L_l$ from $L_p + \sum L_l$ to $L_p$. Likewise, the required number of connections is reduced from $N_{Inj} + \sum L_l \cdot N_{B,l} \sim O(L_p + M + \sum L_l \cdot (N_{F,l} + P_l))$ to $O(L_p + \sum L_l \cdot P_l^*)$, where $L_l \cdot N_{B,l}$ denotes the fact that the binding state complexity per location $N_{B,l}$ needs to be considered for the total number of locations $L_l$ per level $l$ and the multiplication by $P_l^* = (N_{F,l} \cdot K_{p,l} \quad + \quad N_{F,l} \cdot K_{l,p}) < P_l$ denotes the fact that the different levels $l$ now need to connect to the common proxy map $p$ via a local neighborhood that maps their spatial scale and vice versa, by $K_{p,l}$ and $K_{l,p}$, respectively, instead of mapping to each adjacent level. This results in a single proxy map, located in the interfacing module. It serves to encode binding state information and to signal it to NProcs of corresponding positions across the hierarchy and to other brain areas involved in functions such as cognitive control or oculomotor behavior.

## 3. Discussion

Here, we describe a mechanistic model of incremental binding that is able to successfully explain a broad range of experimental evidence and implements the growth-cone hypothesis. The model representations during simulation matched with neural correlates reported in the literature, replicating phenomena of constant up-modulation to signal a task-relevant object, distance-dependent onset of up-modulation [16], and growth-cone-like binding speed variations in presence of distractors [14,17]. Furthermore, lesion studies identified the causal involvement of specific connection types. Direct feedback connections between larger scales of the hierarchical model are necessary to keep the growth-cone-like incremental binding properties intact. Without those connections binding was still able to proceed, but at an

almost constant and lower speed. Beyond these findings, the model is also able to handle more demanding stimulus configurations, such as intersecting curves, and extracts grouped objects that conform with Gestalt laws of visual perception [4].

To achieve these results, the model only needs a single type of neural processors across the hierarchical levels of the visual cortex module and utilizes a common interface for providing output to other nodes in the network. Neurons of the interfacing module filter the activities from the visual cortex module for their binding state by accumulating their output and thresholding it against a high enough value. This threshold mechanism effectively filters visual cortical activities for attended positions of task-relevance. A task module serves as a cognitive control mechanism and provides external input to the interfacing module. This external input takes the form of an attentional seed location and is triggering the incremental binding process between the visual cortex module and the interfacing module.

These design decisions for the model can be contextualized within a theoretical framework of incremental binding. Within this framework we identified constraints and evaluated the individual model components against possible alternatives regarding their connectivity complexity, coding efficiency, and implemented logic. Finally, this complexity analysis of the model and its components showed that the model compares favorably against other viable alternatives for incremental binding architectures. Thus, it makes the proposed model a good candidate for a normative account of incremental binding. To this end, we conclude the section with a list of testable model predictions that arise from this proposal.

## 3.1. Incremental binding could be explained by an interplay of cortex and thalamus

Based on the details of the presented architecture, the question arises how the model's computational structures may relate to visual cortical anatomy. Here, we propose a potential mapping of the computational structures to different areas of the primate brain. The mapping considers underlying model assumptions and their relation to anatomical and functional evidence from neurophysiology. While important aspects are understood already about neural correlates of incremental grouping and (sub-)cortical connectivity schemes in general, much about the involved pathways and structures is still unknown. Thus, the model serves a predictive purpose involving its components across Marr's three levels of analysis (Fig 2).

**3.1.1. Representations in visual cortex are organized hierarchically.** The proposed model is organized according to the structuring principles of the primate visual system. The spatial hierarchy of representational levels best corresponds to visual areas along the ventral pathway [69–71]. Furthermore, the receptive fields of model computational units increase along the model visual cortical hierarchy (cf. Sect 4.1.1; Fig 12) similar to findings about cells in the ventral pathway [61,72,73]. Another aspect are eccentricity-dependent receptive field size effects [74,75] (cf. [76] for an analysis and model). Visual acuity largely depends on the eccentricity of the visual location with respect to the foveal center [77]. The mechanism proposed in this work was studied on a Cartesian grid without eccentricity effects. Thus, we take the model as a first approximation to the processing in central, (para-)foveal regions where such effects would be small and concentrate on the precise mechanisms that facilitate incremental binding and dynamic contextual disambiguation. Further discussion and simulations of eccentricity-dependent effects on the model's processing can be found in the supporting information (S2 Text).

**3.1.2. Perceptual disambiguation happens in parallel across multiple areas.** While the visual cortex is organized hierarchically with respect to its representation, processing among

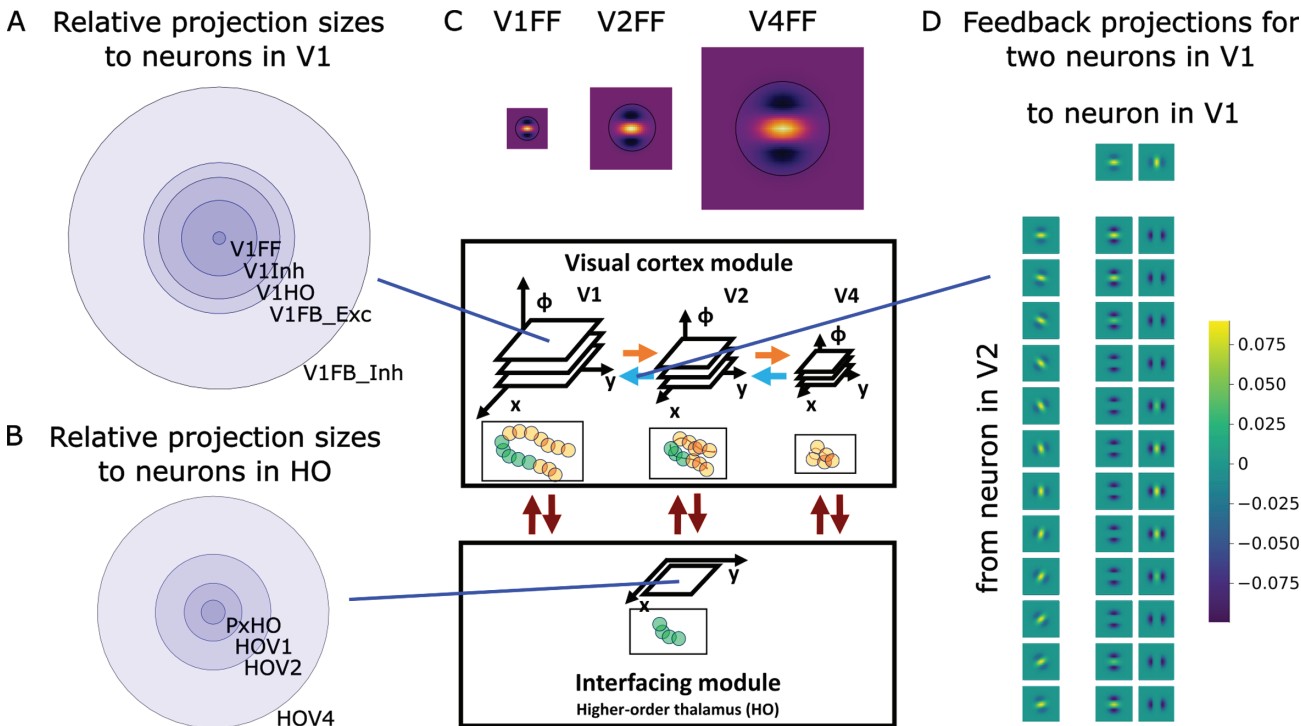

**Fig 12. Overview of relative projection sizes in model cortical areas and higher-order (HO) thalamic nucleus and exemplary feedback kernels.**
(**A**) Relative projection sizes for neurons in model area V1. The reference size consists of the spatial scale from which a feedforward kernel would gather information, i.e., a single grid element in the (x,y)-space of the area (`V1FF`). This feedforward kernel likewise corresponds to the neuron's receptive field. Neurons within each area of the visual cortex module receive projections from different neuron populations: inhibitory surround connections (`V1Inh`), gating connections from the interfacing module (`V1HO`), and feedback projections from model area V2 with excitatory (`V1FB_Exc`) and inhibitory (`V1FB_Inh`) component. Lateral and feedback projections extent over larger diameters than the driving feedforward projections in accord with experimental evidence [61]. (**B**) Relative projection sizes for neurons in the interfacing module. In relation to a single grid element in the (x,y)-space (`PxHO`), model HO neurons receive driving input from model areas V1 to V4 with an increasing projection size (`HOV1` to `HOV4`, respectively). Relative projection size circles are approximations to the real filter sizes (see S1 Text for additional information). A and B are not drawn up to the same scale. (**C**) Approximate receptive field sizes in the visual cortex module. Spatial scales of filters increase along the hierarchy of areas in the visual cortex module. With increasing hierarchical level, the map sizes of areas reduce in spatial resolution and coarsely aggregate information from the higher-resolution lower area (center). Neurons in higher maps effectively achieve a larger integration size by filtering and subsequent downscaling. Upscaling cortico-cortical feedforward kernels between areas onto retinotopic input size provides an approximation of the receptive field sizes (top). By means of surround and feedback interactions actual receptive field sizes might differ from the depiction. (**D**) Exemplary feedback connectivity structure from model area V2 to V1. Depicted are the spatial kernels between two V1 neurons of orthogonal orientation selectivity (top row; horizontal and vertical orientations) and V2 neurons of all orientation channels (left column). V2 neurons of similar orientations provide excitatory feedback if aligned along the direction of preference. Off-axis V2 neurons of any direction provide inhibitory contextual information instead. The colorbar indicates excitatory and inhibitory connection weights of a relative spatial location to the feedback signals. See Fig A in S1 Text for complete feedback connectivity structure.

each of the areas does not happen purely in sequence, but rather in parallel. Once an initial feedforward propagation of stimulus information has taken place, neural activity within each area can develop interdependently on other areas by feedforward and feedback processing [78,79]. Thus, information processing co-evolves concurrently across levels. This ongoing concurrent interaction allows for extraction of stimulus information and computation of contextual information early on, as well as for dynamic disambiguation at later time points. An example of this late-phase disambiguation is seen in the experiments using line crossings in the stimuli (Sect 2.2.4). There, the ambiguous crossing representation evolves early on and only becomes biased towards a horizontal or vertical representation as neighborhood information changes with attentional seed and context (Figs 7 and 8).

**3.1.3. Pyramidal cells integrate different processing streams.** Neural processors within each level represent pyramidal cells and their local inhibitory circuits. The model neurons provide a close functional description of the underlying biological neuron's computation. They capture main findings, namely, apical-basal coincidence detection [60] yielding asymmetric integration of feedforward and feedback signals [80], thalamic gating of respective integration [81,82], and response up-modulation through attention [16]. That pyramidal cells play a dominant role for cortical computation is likewise suggested throughout experimental and computational approaches (see [83,84] for an overview). Since the model's framework provides a clear guidance for the distinct roles of each connection type, one yields interpretable neural representations. A direct mapping can be made between the concept of a binding-state-dependent compatibility computation and the apical integration of the model's neural processor. Thus, the model provides a prediction in assigning such role to the apical dendritic integration of pyramidal cells.

Not all pyramidal neurons are similarly sensitive to attentional up-modulation. Only about half the recorded sub-population of V1 neurons seems to be susceptible to attentional up-modulation (A-units), while the other half is not (N-units; [27,31]). A-units might compose a labeling network that signals attentional relevance given a cognitive state, while N-units might encode sensory information veridically [3,85]. The A-/N-unit distinction gave rise to theoretical considerations about incremental grouping [3,15] and subsequent computational modeling [44,45]. Beyond their mutual correlative properties during incremental grouping [27], it is yet unclear how A- and N-units are interconnected and how they functionally interact. The presented model's pyramidal cell units are similar to A-units leaving N-units beyond the current scope, but without precluding their existence. N-units could hypothetically play role in feedforward filtering and activity normalization to establish the initial base(-grouping) representation. We assume that such more complex feed-forward processing may improve robustness and flexibility (cf. discussion of parameter criticality in S1 Text) but would not qualitatively alter the *algorithmic* model considerations or its complexity requirements to implement the incremental grouping *process* itself. A further strength of the current model description lies in its generic neural model specification, which can describe A- and N-unit behavior by different parametrization of the same set of equations (cf. discussion in [86]). This way it is as well conceivable to model a population of neurons with a gradual distribution of A- and N-unit-like properties.

**3.1.4. Higher-order thalamus reciprocally interacts with cortical areas.** Another important component of the model consists in the interfacing module. It implements a neural sheet of proxy nodes, serves as an interface between hierarchical levels of the visual cortex module, and gates the apical-basal integration of model pyramidal cells. We suggest that this proxy node computation is best matched by higher-order proportions of the visual thalamus, such as the pulvinar. This suggestion is supported by experimental evidence as discussed below. Still, our suggestion poses a model prediction as no distinct measurements for causal thalamo-cortical interactions during incremental binding exist to date.

Anatomically, higher-order thalamus forms direct connections with several cortical areas along the visual pathways [87,88]. Given these many direct connections, thalamus and cortical areas are reciprocally connected in a shallow hierarchy [62]. There, visual cortical areas are predominantly connected to the pulvinar [89] forming overlapping projection regions [88,90] of which at least two are retinotopically organized [91–95]. The presented interfacing module likewise establishes a shallow hierarchy by direct connections with the different model visual areas. This allows for fast communication between areas which would be otherwise

considered distant in a hierarchical view, such as V1 and the task module (Fig 3). Importantly, this allows for attentional selection and up-modulation to align at roughly the same onset latencies, as measured during visual curve tracing [32].

Functionally, pulvinar regions are involved in processing of visual stimuli [96]. Findings indicate roles in processing visual salience [97], space- and feature-properties [98–100], action formation [101–105], and attentional processing [106–108] and selection [109,110]. More generally, the pulvinar was found to gate cortico-cortical information transmission [106,111], and, likewise, pulvinar representations seem to be gated by cortical attentional influences [112]. Such mutual exchange of attentional signals resembles the interaction of the proposed interface module in the model with the model visual cortical areas (e.g., Fig 4B). Whether the thalamic representation is rather featureless or coding for specific features, like visual motion or motor information [99,100], is not completely clear yet. But, overall, the thalamic representation seems to be involved in task-related processing [109,110] and is lower-dimensional than that of the associated visual cortical areas. These properties are likewise crucial factors of the proposed interfacing module.

Inactivation of the pulvinar has been found to affect performance in attention paradigms [113] and filtering of visual distractors [114–116] up to a point that area V4 is thought to enter an "inactive state" during visual processing of stimulus changes [107]. Pulvinar inactivation furthermore affects cortico-cortical interactions showing that a causal relationship between pulvinar and visual (attentional) processing exists [117]. Such coordination of attention and task-dependence might be relayed through pulvinar from frontal cortical regions forming a fronto-parietal network [118,119]. At the level of pyramidal cells in cortex recent neurophysiological evidence suggests a gating role of thalamic input sources for apical-basal integration [82]. We modeled the thalamic control on the apical-basal integration process through direct coupling onto the pyramidal cell apical trunk. A more detailed implementation might mediate such control indirectly via apical (dis-)inhibitory circuitry [120–123]. Such a dis-inhibitory circuit could then also account for further pathway-specific gating [124] or aspects of thalamo-cortical interaction in synaptic long-term adaptation [125]. Similarly to pulvinar inactivation [117], the model lesion experiments indicate a causal role of the interfacing module in attentional processing (Fig 6).

**3.1.5. Higher-order thalamus mediates phases of attentional processing.**   At the computational level, several theoretical frameworks have been proposed which are based on the experimental findings discussed above. Adaptive circuitry in pulvinar could support functions of visual cognition [126] and might mirror the ventral-dorsal processing pattern of visual cortex [127]. Cyclic (dis-)engagement of attention coordinated via thalamic nuclei led to the proposal of a rhythmic theory of attention [128]. There, theta-band oscillations coordinate phases of attentional engagement with visual stimuli (sampling, exploitation) and attentional disengagement to shift attention between processing sites (shifting, exploration) [129]. In line with this theory, behavioral performance was shown to fluctuate in a theta rhythm [130–132]. During complex tasks also other frequency bands may be important, such as the alpha-band during memory tasks [133]. The role in leading and following the theta-band oscillation shifts between the fronto-pulvino-parietal network depending on the respective processing phase [128]. In the proposed model several modules likewise interact via the interfacing module to exchange information for attentional processing of stimuli. There, rhythmic phases can be identified during which a module provides input to the interface or receives output from it. For example, the task module first provides an attentional seed as input to the interface and triggers the binding process, while later on it can receive output from the interface in terms of the final tracing outcome (Fig 4C).

Coherence of theta-band oscillations could establish functional networks between cortical areas [134] to monitor and coordinate attentional task demands [135]. This coherence may be mediated by higher-order thalamus [136] with waxing and waning theta-band power dependent on attentional engagement [119,137] and may be reduced under prolonged attentional exploitation [138,139]. How brain-wide frequency-band specific oscillations relate specifically to the attentional setting encountered in incremental grouping has not been specifically investigated yet. We propose that the model simulations provide an idealized view on the attentional spreading mechanism under sustained, on-going attention, where theta-band activity is low and frontal and occipital visual regions interact coherently [139]. There, the interfacing module concurrently communicates incremental binding states to the model visual cortex areas and the task module to keep them in sync. Once a perceptual decision is reached, the task module might alter the state of the interfacing module and shift the seed of attention to a new position to start another cycle of attentional processing.

**3.1.6. Thalamic spiking and attentional processing.** While the presented interfacing module employs rate-based neurons, thalamic neurons possess response properties that allow for interesting predictions about a spike-based model extension. Thalamic neurons across different sub-divisions exhibit two distinct modes of firing [140–142], namely tonic and bursting mode firing, with relations to slow-wave sleep states [143], activity synchronization [144] and sensory information processing [145], such as detection versus discrimination [140,146]. The firing mode depends on hyperpolarization from inhibition [147] where prolonged processing periods and adaptation to sensory input result in tonic firing regimes [140,148], while the first stimulus response is oftentimes a thalamic burst [145,149,150]. Furthermore, thalamic bursting has been shown to reduce cortical theta-band coherence [151].

A spike-based version of our thalamic model would have a clear relation between tonic-vs.-burst mode firing and theta-band-related phases of attentional (dis-)engagement and processing (Section 3.1.5). Thalamic processing sites under prolonged attentional engagement would reside in a tonic firing mode and synchronize with respective cortical neurons in a theta-band regime. Thalamic neurons at unattended or actively ignored processing sites may become less active due to inhibitory influences. This inhibition may be lifted during attentional disengagement and sampling. Post inhibition, thalamic neurons will reside in burst mode and yield higher susceptibility to cortical sensory evidence. Therefore, previously unattended sites will have an elevated probability of being detected and selected by the prefrontal task module for the subsequent phase of attentional engagement and tonic-mode firing.

**3.1.7. Cognitive theories rely on thalamo-cortical interactions.** From the cognitive processing perspective, the role that was assigned to the thalamus has changed in light of evidence from the past decades. From being a mere relay station that duplicates (sub-)cortical signals its role has now become a more important one involved in coordination of cortical signaling [152,153]. Yet, its precise functional role is still debated and led to a range of (non-)exclusive suggestions [154]. Suggestions closest to the original relay station idea assign the thalamus the role of a *gatekeeper* [155,156] or *switchboard* [157] selectively (de-)activating trans-thalamic signals. Similar to this is the idea of the thalamus as a *hub* [158–160] integrating different signal streams and promoting them to other brain regions for different cognitive functions. Focusing on the integrating behavior of the thalamus, but for more specific cortical functions, it has been suggested that thalamus serves as (part of) a *blackboard* [161,162]. There, cortical processes can write information to the thalamus to store and update information, and read from it to communicate with each other. In this regard, thalamus would store current system state information and provide a buffer that exposes a common interface

to different cortical areas which take part in executing a common function. A similar black-board role has been suggested for primary visual cortex [163,164] opening up the possibility that thalamus and visual cortex jointly implement a blackboard. Focusing rather on behavioral implications, the thalamus could play a *task-related role* as well (cf. Section 3.1.4). In a broader view of a hub that integrates processes across cortical sites it is also suggested that thalamus serves as a *gateway* for permitting information and regions into a conscious state [165–170]. In a recent review Cortes et al. discuss the role of the pulvinar in global cortical brain networks for visual processing further [170]. In this review they identify the pulvinar as a structure modulating cortico-cortical signaling and provide links to the global neuronal workspace (GNW) theory of conscious processing and predicitve processing paradigms. In these paradigms the pulvinar may serve the role of establishing an integrated state of processing by enhancing cortical processing, e.g., enhancing coincident cortical inputs to an area during the ignition phase of the GNW, or regulating the transmission of (un-)reliable cortico-cortical information in predictive inference processing.

In our model the thalamic regions are modeled by the interfacing module and are assigned properties of task-related coding, interfacing, and blackboard functionality. In terms of a *task-related role*, it aggregates coarse-grained location-specific information about task-relevant integrated representations from cortical pyramidal cells. In terms of an *interface*, it can be used as such for attentional control, as the task module can provide a seed location of attention to the visual cortex module via the interfacing module's proxy nodes. As a *blackboard*, the interfacing module contains the binding state of the visual cortex module, aggregates it in a map, and provides supra-threshold activity for those positions that signify binding states. As a *hub* or *gateway*, the interfacing module projects this supra-threshold activity back to the visual cortex module to enable cortico-cortical feedback integration. This aspect of the interfacing module is similar to the proposal of Cortes et al. in that it helps to build an integrated global network representation [170]. The predictive processing aspect of the proposal from Cortes et al. may seem in stark contrast with our model at first, in that predictive processing is oftentimes associated with subtractive mechanisms and activity reduction, while the proposed model is built on cortical activity enhancement through thalamo-cortical interactions. In fact, these two views can be reconciled since predictive processing by mechanisms of enhancement are more likely candidates for realizations in the brain [123]. Overall, the interfacing module takes part actively in the binding process and exerts cognitive control onto the cortical architecture based on task-related guidance signals in line with existing evidence.

**3.1.8. The multiple sites of attentional processing.** In our model the thalamic nucleus serves as an interface for attentional signaling between the visual cortex module and other modules of attentional control. These other modules are abstractly subsumed as the *task module*. For the strict functioning of the presented simulations, it would not be necessary to factor out a separate task module from the interfacing module. Yet, this segregation highlights the role and flexible attribution of the source of attentional processing. Such task module provides attentional control to the visual cortex module arising *externally* to the interfacing module but communicates this signal *through* the interfacing module.

Where such task-related signals are generated before they are passed to the interfacing module, and whether these sites represent the locus of attention in a similar spatial map, is beyond the scope of the model. Many brain regions and sub-systems have been reported to be involved in attention, but whether each plays a causal role is not yet clear [171,172]. Several cortical regions are plausible candidates representing such locus of attention. Candidate regions are prefrontal regions, such as the frontal eye fields (FEF) [32,173,174], posterior regions, such as intraparietal sulcus (IPS) [175], or also temporal areas [176], such as the

dorsal part of the posterior inferotemporal cortex (PITd) [177]. Additionally, other subcortical sites, such as the SC (Superior Colliculus), further modulate this processing by thalamic projections [178,179].

While above evidence suggests that especially higher-order thalamic sites play a relevant role in cortical attentive function, it nevertheless cannot be ruled out, that the role of the model's interfacing module is not representing higher-order thalamic sites. Instead, other cortical areas could directly function as an interfacing module with likely candidates from the above being PITd [177] or IPS [175]. In this case, the specific connectivity pattern, as implied by the model, is rather unlikely. There, the candidate cortical area would need to connect to the apical dendrites of each pyramidal cell in the visual areas and subserve a specific gating role. Such gating signal would need to be distinct from the contextual evidence aggregation communicated by other apical feedback input, and therefore, would likely require a different kind of connection type to selectively impact the computation.

Thus, given our findings, we suggest that incremental binding might best be explained by an interplay between cortex and higher-order thalamus.

## 3.2. Broader implications of the complexity considerations

The investigations about the complexity of different computational motifs for incremental binding and their connectivity patterns serve a broader purpose beyond judging the proposed model's complexity. Our assumptions and constraints extend an earlier framework for algorithmic descriptions of incremental binding [13,15]. The framework allows to reason about incremental binding on an algorithmic level and our complexity considerations can help in guiding further modeling constrained by empirical as well as computational restrictions. Furthermore, existing models can be analyzed with respect to design decisions (Fig 10B) and how well they fit a normative account regarding their coding and the representational or connectivity complexity. Dependent on the model these decisions are either made explicitly or implicitly. When it comes to coding for the different states of a node, different choices are possible. These choices can be, e.g., the introduction of a separate population and/or channels for the incremental representation versus the base representation [43–45,47–49,180]. Likewise, a multi-step approach could be a solution to provide distinct phases of base evidence accumulation vs. incremental binding propagation [10,42]. Furthermore, existing learning-based architectures, for which the relationship between different pathways and neurons in the network and their representational role might be less clear, could be analyzed in this regard [50,65,181,182].

Unlike the previously mentioned models, the currently proposed model does not foresee completely separate populations for the base representation and incremental binding state, nor does it require phasic processing. Instead, it utilizes neural processors with basal and apical integration sites and a by-pass route via the interfacing module, that specifically filters for binding state information (Fig 3).

Investigating all the different models further with the tools used here, e.g., regarding computational and representational complexity, could provide further insights into the conceptual strengths and weaknesses of the respective models. Likewise, these comparisons could provide further insights into which model choices lend themselves for easy trainability within (deep) learning frameworks and shed light on why that might be the case. Also, models become comparable based on quantities of representational and computational complexity. There, the implementational overhead per model can be estimated in relation to an ideal realization that optimally incorporates the outlined complexity considerations.

The design space laid down by our considerations offers additional modeling solutions, which have been unexplored so far. For example, the same neural processor could code for gradual base evidence via tonic firing rates and switch into a state propagating binding information by providing higher-rate bursting output with a distinct temporal signature. There, different recipient nodes could filter the neuron's output for low- vs. high-frequency codes to provide multiplexing among a single neuron's output channel (one channel, ternary NProc solution, Fig 10B; [183]).

### 3.3. The broader context of binding and visual-cognitive architectures

The investigated architecture and proposed framework focused on incremental binding mechanisms in visual grouping. Nevertheless, one of the underlying assumptions for all the considerations has been that visual cortex is capable of flexible processing dependent on task demands and attention. Thus, it should not only be able to perform incremental binding, but can engage in different visual functions, such as grouping, searching, comparing, or tracking objects.

This generality and flexibility have been discussed in the context of visual routines [8,9] and cognitive programs [10]. The common idea is that, similar to a computer, visual cortex can make use of different encapsulated computational motifs to find solutions to specific tasks, and that more complex tasks can be tackled by compositions of these simpler motifs. The model proposed here implements one such motif, namely incremental binding. As an elemental operation, it therefore needs to be able to read data, e.g. from the task-module, store data in the interfacing module, and perform recursive binding starting with the attentional seed and ending when no further increment to the grouped object is encountered. What is missing so far in terms of visual routines and cognitive programs for the proposed model is the flexible combination with other operations and a task-dependent selection of the respective operations. Nevertheless, the proposed model provides a clear interface for possible interaction with a cognitive control mechanism for exchange of intermediate results with other operations and does not possess any hard-coded components that would restrict its function purely to incremental binding.

To realize the elemental operation of incremental binding the system is evolving in a decentralized and parallel fashion. Yet, the incremental binding process unfolds only sequentially in time and localized in space. It depends on the availability of local neighbors for compatibility computations and on a gating process that is likewise dependent on a local neighborhood. This parallel-yet-sequential processing has similarities to earlier work [3,6,7,184], where decentralized processes operate in parallel until a sequential bottleneck forms a decision, that then can lead to the next step of a parallel execution process. For the case described here, this formation of a rather discrete decision (bound vs. not bound) is happening by the accumulation process within the interfacing module. It accumulates the evidence for a binding state at the respective position across a decentralized pool of neurons in the visual cortex module. Dependent on the state of this accumulation the interfacing module then gates the parallel and decentralized apical-basal integration processes of the pools of neurons among the hierarchy of the visual cortex module.

On a broader scope, the model is compatible with theories of cognitive architectures. For example, the model's interfacing module could be interpreted as a buffer of visual information in terms of ACT-R [185]. Likewise, the flow of task information in the model is compatible with considering it as a parietal-prefrontal processing sub-system [186]. In such framework, a task-module first triggers a computational motif in the visual cortex module and later on reads out the resulting attended object.

Relating the model to other theories of binding and object-based attention commonalities and differences can be found. In comparison to the feature integration theory (FIT) [53], the model would not represent specific stimulus features, such as contour or color in a decomposed fashion, rather they would be assigned to a common high-dimensional feature space in the visual cortex module, where different neurons would be differently selective to different features. This distinction from a decomposed representation of feature spaces only will become clear in more sophisticated model versions, since so far, the model was concerned with processing only a single feature dimension, i.e., oriented contrast information. A component in common with FIT is the existence of a spatial relevance map that helps guiding the selection of what places should be bound. In the proposed model this role is assigned to the task module, which provides attentional selection to the other modules of the model. In comparison to a more recent conceptual proposal for an architecture of attention [59], the model's interfacing module possesses commonalities with the idea of gain maps. Gain maps interact with a hierarchically organized stream of levels that perform feedforward-feedback interactions. There, mutual projections between hierarchical levels and gain maps selectively influence the respective hierarchical processing in favor of the maps' contents. This motif of selectively biasing processing is also found in the model's interfacing module. It does so by representing a spatial projection of binding state information from the visual cortex module and back-projecting this information onto a local neighborhood across the hierarchy. Yet, the interfacing module is mostly concerned with mediating this process of selection and could likewise be influenced from other gain-map like areas that influence the selection process further (cf. Sect 3.1.8).

## 3.4. Model predictions

Based on the modeling contributions with operations on multiple levels of analysis and its inspiration by experimental evidence, several model predictions can be derived:

- Incremental grouping happens by a state of coincidence detection leading to response amplification within pyramidal cells [60]. Thus, processes gating this apical-basal integration play a crucial role in incremental binding. As a consequence, attentional visual binding, and therefore incremental grouping, is causally dependent on gating signals from higher-order thalamus [82]. This implies relations to the states of (un-)consciousness under which the incremental binding operation could occur [169]. To completely understand this interfacing and gating it may be necessary to extend investigations from a single nucleus to a network of thalamic nuclei instead. Yet we suggest that the first region to test would be the pulvinar given the mentioned evidence (Sections 3.1.4 and 3.1.5).
- Growth-cone-like tracing speed variations are established by grouping within different visual areas at different resolutions. These grouping processes are influenced by spatial context information that is transmitted by feedback projections to apical compartments of pyramidal cells. Deactivating such contextual feedback projections between higher-level areas would require tracing to only rely on lower areas. These lower areas receive more localized feedback from intermediate levels resulting in slower tracing speeds (Fig 6). That this tracing happens depends on the coincidence of feedback with feedforward information. For simple, solid line stimuli this feedforward information has been extracted by a base grouping process. How tracing would extend to illusory contours remains an open question. It will require to reveal which base-grouping processes establish illusory contour representations and where along the visual hierarchy they are

established. For example, incremental grouping could then require intermediate-level cortical areas beyond V1 to access base representations.

- Incremental binding is a generic mechanism that acts similarly upon different representations dependent on task demands. The prime examples are incremental grouping of contour or surface elements. As a result, the process is highly decentralized, but an integrated functional network can be formed by apical contextual signaling and cortico-thalamo-cortical interaction. Even if our prediction about the role of higher-order thalamus in this computation would be falsified, the complexity analysis still predicts structuring principles for the algorithmic and implementational level. For example, the analysis predicts that there should exist lower-dimensional integration nodes, or hubs, that serve to reduce dimensionality and therefore connectivity complexity. This impact on the complexity of the binding operation becomes even more tremendous if it acts upon a high-dimensional base representation, as required for real-world scenes. If such a dimensionality reduction principle is at work, the complexity analysis predicts additionally, that the attentional binding should be a compound of a context-and-feature-specific and an attention-dependent feature-unspecific processing stream represented at distinct computational nodes of the underlying neural network (Fig 10D and 10E).

The above predictions outline incremental binding as a multi-scale multi-area problem operating in a context dependent fashion. Thus, to understand the underlying mechanisms thoroughly, further experimental and theoretical investigations need to be conducted. For these investigations the above model predictions provide guidance to better understand and constrain the exploration space.

## 3.5. Limitations and possible extensions

**3.5.1. Solving more complex visual tracing problems.** The framework and model have been developed to investigate incremental binding processes operating on oriented contour segments. It has also been tested on additional, more demanding tracing configurations, such as broken lines [5], curvatures [35], clutter [5], and intersections [38]. Further stimulus configurations have been reported in the literature, but were not considered here. These incorporate additional features, such as color, or other elements as line components, e.g., Gabor patches [5]. Additionally, the experimental evidence covers surface- and object-based cases of incremental grouping [14,34,40,41]. Overall, scaling the proposed model to such tracing stimuli will require an extension of the base representation's feature space.

The range of model simulations shows that the model already covers a broad range of stimuli from the literature (cf. Section 2 and S2 Text). Likewise, it becomes apparent that for some of the configurations additional feature space components would be required to yield more robust and generalizable simulation results. For example, grouping currently becomes impossible once the element spacing of dashed lines increases beyond a certain limit or if distractors are placed too close by. Incorporating additional base grouping mechanisms, such as long-range filters [187–189], could therefore enrich the representation. Furthermore, the model is able to incrementally group contour elements in the presence of crossing contours relying on Gestalt properties of good continuation, but does not work properly if lines meet under acute angles (Sections 2.2.4 and S2 Text). A possible solution would be to extend the representation by introducing end-stop-, junction-, or curvature-selective neurons [44,45,190, 191]. Incorporating curvature-selective units could further enrich the model simulations to make tracing speeds curvature-dependent, and, thus, more similar to experimental evidence ([35]; cf. Section 2.2.5). An extension to surface-based incremental grouping processes [14],

likely requires incorporating elements of surface-coding and border ownership computation [192,193].

Another example of a more complex interplay between features consists in a version of the curve tracing task which is based on motion coherence [20]. While a model for this task exists [194], it remains unspecific with respect to the incremental versus parallel nature of binding and invites for further investigations using the model proposed here with a base representation extended for motion processing.

Notably, all the additional representational elements mentioned above would lead to enhanced base grouping and a richer base representation. Upon this parallel processing, the proposed incremental mechanism would still perform the sequential spreading as proposed, given that the compatibilities in the enriched feature space are correctly expressed in terms feedback projections.

**3.5.2. Model extension to realistic visual scenes.** Another interesting direction consists of extending the framework beyond incremental grouping displays to more realistic feature spaces and object(-part)-level representations [11]. Real-world images exhibit high-dimensional feature spaces with intricate relations between features, such as color, texture, and shape. Ambiguities may only be resolved by incorporating higher-level semantic context, for example, what *kind* of action a person is performing with an ambiguous object. Such information provides selective context about which locations of a scene to group to the segmented object. A first step to incorporate such semantic context may consist in trying to flexibly connect the proposed model with object category information extracted by another model for object processing (cf. [195] for an overview). We predict that the interaction skeleton expressing compatibilities of spatial features with these semantic elements will become more complex, but that the incremental binding *mechanism* operating upon this skeleton will stay the same. Thus, the proposed model's incremental binding mechanism should be extensible to more complex configurations and ultimately be scalable to real-world scenarios.

**3.5.3. Learning the incremental binding operation.** Scaling the proposed model to realistic feature spaces will make handcrafting of filters prohibitive. Therefore, it will be useful to further investigate the proposed model in the context of learning an artificial neural network (NN) architecture, for example, adapting a deep NN or deep recurrent NN. Such NN model architecture can then be trained and evaluated on contextual binding tasks [196] or segmentation challenges [42,65,197]. There, it can then be compared to other deep learning models with relation to incremental binding that have either been investigated on distinct contour image data [47–49,65], or image segmentation datasets [42]. Exemplary results on the pathfinder dataset [65,66] indicate that the presented model is generally able to solve such segmentation challenges (Fig 9F and 9G), but learning to solve the complete dataset is left for future work.

The task of the pathfinder challenge is to detect the connectedness of two dots by target line elements placed within a field of similar distractors [65]. The network proposed alongside the dataset uses the newly introduced horizontally gated recurrent unit (hGRU) as a building block to solve the task. A comparison with our model reveals similarities and differences that open up avenues for future investigations. In the hGRU model propagation of contour grouping information happens within the hGRU block after input transformation by a convolutional layer, which is similar to providing a base representation for a subsequent incremental process. The hGRU unit consists of two interacting states, which are motivated from dynamical neuron descriptions of interacting excitatory and inhibitory cells, similar to our model. Yet, for the hGRU model it is not clear from the state interactions, how the grouping arises on

a mechanistic level. Synthesizing deep neural network building blocks from the model proposed here may be able to yield such a mechanistic explanation. In comparison to the hGRU model that computes grouping on a single scale, our model would consist of a hierarchy of interacting recurrent units (Fig 3B) at different spatial scales (model visual areas) and a gating mechanism connected in parallel (interfacing module; Fig 3A). While training such network might be a challenge, similar recurrent units have been trained successfully already on simplified tasks using backpropagation through time [198]. Also, further promising approaches exist, such as RELEARNN [47–49] or C-RBP [65], to eventually train the proposed model.

**3.5.4. Understanding thalamo-cortical interaction.**   Regarding the modeling of thalamic function in thalamo-cortical interactions, relations to other existing relevant models would be of interest. Logiaco et al. show how exerting control over cortical computational motifs can be efficiently realized by smaller populations of thalamic neurons [199]. There, ensembles of less detailed model neurons and the connection density are the guiding principles. Munn et al. show how the formation of state-dependent pyramidal cell bursting can be linked to thalamo-cortical interaction [200]. They provide an information-theoretic investigation and link the thalamo-cortical interaction to a maximization of integrated information in pyramidal cell computation. Remaining on a finer physiological level, the simulations, yet do not imply a specific algorithmic function, but rather provide an argument for optimized information transmission. In a related large-scale model for more abstractly investigating attractor dynamics of thalamo-cortical interactions, Müller et al. show how thalamus can play an important role in shaping the cortical information state regarding consciousness [201]. Likewise, a joint analysis-and-modeling study of magnetoencephalography (MEG) data indicates the important role of cortico-thalamo-cortical interactions in changing functional network connectivity during stages of a cognitive task [202]. Investigating interrelations between these models and the work presented here could further the understanding of thalamo-cortical interactions.

**3.5.5. Further complexity analyses.**   Further insights could be obtained by challenging the complexity analysis (Section 2.3) and existing models concerning, e.g., the growth cone hypothesis. For example, the analyzed attentional seed injection methods investigated in Section 2.3.4 identified a shortcoming for direct top-injection. Interference cases between the target object's representation and a nearby distractor renders this case ineffective prohibiting backprojection of the binding signal. Yet, a solution could exist if, e.g., the investigated feature space becomes high-dimensional enough, such that the interference is resolved simply by the sparsity of the high-dimensional embedding. Testing such ideas also experimentally might shed further light on how the brain implements incremental binding on a neural level. Additionally, the proposed framework could be used to consider how easy or hard it is to learn specific required connectivity structures (Fig 10B) and parameter combinations for different models to judge their biological plausibility.

Furthermore, the proposed model has been designed keeping in mind the necessity of providing an efficient and flexible interface for visual operations. Thus, the architecture and the theoretical complexity analysis could be used to investigate visual search. Visual search is commonly investigated by theories that try to explain binding in the brain [1, 203,204], and therefore constitutes an interesting target for further model investigations. A resulting model may then be able to implement both, incremental binding, and visual search.

## 4. Materials and methods

### 4.1. Model details

**4.1.1. Network model.** On a network level, our proposed model can be described as a graph of nodes (neural processors) and channels (directed connection types). Nodes are arranged in retinotopic space and code for certain input features, i.e., orientations, forming feature maps of a model area. The areas of the visual cortex module form a scale space and have a relative spatial size with respect to the input of $[1, 0.5, 0.25]$ for the areas V1, V2, and V4, respectively (Fig 12; parameters $s^V$ in Table A in S1 Text). This scale space results in smaller maps of reduced spatial resolution for higher levels of the hierarchy (cf. Figs 4A and 7B). The interfacing module likewise forms a retinotopic map, but only with a single feature channel, i.e., the binding state, and a relative size of $s^{HO} = 0.66$ ranging between the spatial resolution of V1 and V2. The channels connecting the neural processors are modeled by localized interaction kernels. Kernels abstractly represent underlying neural connectivity patterns and provide weighted connections to nodes in a local space-feature neighborhood around the target position (cf. Fig 12D for feedback connectivity; cf. Fig A in S1 Text for a complete illustration). The neighborhood size depends on the connection type, i.e., driving feedforward, inhibitory surround, gating input, and feedback (Fig 12A and 12B; [61]). Updates among these connections are propagated by computing the convolution of the respective kernel with the source nodes. Overall, the application of a kernel is performed by

$$g^{Trgt,Src}(x_i, y_i, \theta_i) = f\big(resize\big(\big(\Lambda^C_{\theta_i} * r^{Src}\big)(x, y, \theta), s^{Trgt}\big)\big). \tag{1}$$

Each node has a certain retinotopic position $x_i, y_i$ and a feature value $\theta_i$ it is coding for. Here, $\Lambda^C_{\theta_i}$ describes the kernel of channel type $C$ that represents channels connecting the source node $Src$ with the target node $Trgt$ based on the source node's activity level $r^{Src}$. The $*$ operator denotes a spatial convolution with circular boundary treatment among the feature dimension and zero-padding among the spatial dimensions. A resizing operation $resize(\bullet, s^{Trgt})$ is applied after each convolution using linear interpolation in space, such that the filter response "$\bullet$" matches the target map's spatial size. Joint filtering and resizing effectively implement a scale space, where down-sized maps span larger input distances at lower spatial resolution (Fig 12C). Thus, when cortico-cortical projection kernel sizes are measured in terms of neurons covered, a rather constant parametrization can be utilized across the different scales of the hierarchy (cf. S1 Text). When kernels are expressed in terms receptive fields in input space the hierarchical scale space becomes apparent (Fig 12C). To restrict the range of input values to each node, the convolution operation is followed by a non-linearity $f(x) = min(max(x, 0), 1)$ that realizes a rectified linear response with saturation ceiling level.

The overall connectivity scheme of the network is given by Table 1. Within the network, four types of channels exist: feedforward (FF), feedback (FB), and proxy-related, with a distinction into channels projecting to (B1) and channels projecting from (B2) the thalamic interfacing module. The local pyramidal cell processing of a neighborhood is aided by an inhibitory (Inh) interaction.

The respective kernels between nodes of the hierarchical levels are space-feature separable and defined by

$$\Lambda^C(x, y, \theta) = \Lambda^C(x, y) \cdot \Lambda^C(\theta), \tag{2}$$

where $\Lambda^C(x, y)$ defines the kernel component in retinotopic space and $\Lambda^C(\theta)$ the kernel along the feature dimension. There, kernels of the feedforward channel $C = FF$ consist of a Gabor

**Table 1. Network connectivity structure. See text for details.**

| To<br>From | Inp | V1 | V2 | V4 | HO | S |
|---|---|---|---|---|---|---|
| Inp | - | FF | - | - | - | - |
| V1 | - | Inh | FF | - | B1 | - |
| V2 | - | FB | Inh | FF | B1 | - |
| V4 | - | - | FB | Inh | B1 | - |
| HO | - | B2 | B2 | B2 | - | - |
| S | - | - | - | - | B1 | - |

kernel for the spatial component

$$\Lambda_{\theta_i}^{FF}(x,y) = Gabor_{\sigma,\lambda,\theta_i}(x,y), \tag{3}$$

with a standard deviation $\sigma$ for its envelope, wavelength $\lambda = 2 \cdot \pi \cdot \sigma$, and orientation selectivity $\theta_i$, and a Dirac kernel along the feature component

$$\Lambda_{\theta_i}^{FF}(\theta) = Dirac_{\theta_i}(\theta), \tag{4}$$

for a given target node's orientation $\theta_i$.

Kernels of inhibitory type $C = Inh$ consist of a Gaussian kernel along the spatial dimension

$$\Lambda^{Inh}(x,y) = Gauss_\sigma(x,y), \tag{5}$$

with standard deviation $\sigma$ and a uniform kernel along the feature dimension spanning all feature channels, i.e., size $\rho_\theta = n_\theta$,

$$\Lambda^{Inh}(\theta) = Uniform_\rho(\theta). \tag{6}$$

The composite inhibitory kernels have been normalized to a sum of 1.

Kernels of feedback type $C = FB$ consist of a Gabor kernel along the spatial dimension (Eq 3) and two feature kernels that are separately convolved with the positive and negative values of the Gabor filter respectively (Eq 2). While the feature kernel for positive values is given by a rectified cosine half-wave

$$\Lambda_{\theta_i,+}^{FB}(\theta) = max\big(cos_{\theta_i}(\theta), 0\big), \tag{7}$$

the feature kernel for negative values $\Lambda_{\theta_i,-}^{FB}(\theta)$ is a uniform kernel (Eq 6). The feedback conductance is computed by filtering the input from the next higher level with each of the kernels $\Lambda_{\theta_i,+}^{FB}(x,y,\theta)$ and $\Lambda_{\theta_i,-}^{FB}(x,y,\theta)$ separately and afterwards adding up the result:

$$g^{Trgt,FB}(x_i,y_i,\theta_i) = g_+^{Trgt,Src}(x_i,y_i,\theta_i) + \xi \cdot g_-^{Trgt,Src}(x_i,y_i,\theta_i), \tag{8}$$

where $\xi$ is a scaling factor for the negative contribution of the feedback signal. Gabor filters for both, *FF* and *FB* connections, had a phase shift of 0 (even component), an aspect ratio of 1, and had their positive and negative values normalized to sums of 1 to compensate for the DC component. To avoid cut-off artifacts matrix sizes of the Gaussian and Gabor kernels were set to $1 + 2 \cdot 5\sigma$.

The kernels connecting the hierarchical levels of the visual cortex module with the interfacing module are defined as follows. Binding channel connections of type $C = B1$ connecting a hierarchical level source $Src \in \{V1, V2, V4\}$ to the interfacing module $Trgt = HO$ are given by a composite kernel (Eq 2), where the spatial component, as well as the feature component are defined by uniform kernels (Eq 6; with limited extent in space $\rho_x \times \rho_x$ and spanning all feature channels $\rho_\theta = n_\theta$). The binding channel connections of type $C = B2$ connecting the interfacing module $Src = HO$ to a hierarchical level target are purely spatial kernels. Remember that the interfacing module does not code for any feature $\theta$ and the connections are likewise defined by uniform kernels (normalized to sum of 0.5) with limited spatial extent (Eq 6).

**4.1.2. Pyramidal cell model.** Within the network each node represents a neural processor that abstractly captures the computation of a local neural microcircuit of finer detail. For the hierarchical levels' nodes, this microcircuit can be represented by a cortical column model [205]. We describe its computation by means of a pyramidal cell, which is thought to be the most relevant processing element of such microcircuits [83,84]. The pyramidal cell model consists of two dynamical equations describing the temporal evolution of basal $b$ and apical $a$ membrane potentials:

$$\tau_b^V \dot{v}_b^V = -v_b^V \cdot g_b^{V,Leak} + (\beta_b^V - v_b^V) \cdot g_b^{V,Exc} - \kappa_b^V \cdot v_b^V \cdot h_b^{V,Inh}, \qquad V \in \{V1, V2, V4\}, \qquad (9)$$

$$\tau_a^V \dot{v}_a^V = -v_a^V \cdot g_a^{V,Leak} + (\beta_a^V - v_a^V) \cdot g_a^{V,FB}, \qquad V \in \{V1, V2, V4\}. \qquad (10)$$

Such cells reside on all levels $V$ of the model hierarchy, i.e., areas V1, V2 and V4. These conductance-type equations capture the integration of basal, peri-somatic influences on the cell's membrane potential and the changes on apical distal dendritic potentials [60]. Here, constants $\beta^V$ describe the excitatory reversal potentials, which provide an upper bound to the membrane potential values, $\kappa^V$ is a scaling constant for the divisive inhibition, and constants $\tau^V$ steer the time scales of the temporal evolution of the state variables. Conductances $g^{V,T}$ describe input-dependent influences on the potential based on specific input channel types $C \in \{FF, Inh, FB, B2\}$. The inhibitory component $g^{V,Inh}$ is mediated by a non-linearity $f^{Inh}(x) = 1.3/(1 + \exp(-(x - 0.25) \cdot 20))$, which captures the response behavior of local inhibitory interneurons

$$h^{V,Inh} = f^{Inh}\left(g^{V,Inh}\right). \qquad (11)$$

Additionally, $g^{V,Leak}$ describes a constant leak, and the excitatorily driving conductance $g^{V,Exc}$ is defined by

$$g_b^{V,Exc} = k^{V,FF} \cdot g^{V,FF} \cdot \left(1 + \lambda^V \cdot g^{V,B2} \cdot h_a^V\right). \qquad (12)$$

This equation describes the three-way interaction between the feedforward channel, the feedback channel, and the interfacing module input, where $k^{V,FF}$ and $\lambda^V$ are scaling constants. The feedback channel's influence is mediated by the apical component via

$$h_a^V = max\left(v_a^V, 0\right). \qquad (13)$$

The three-way interaction among these inputs is asymmetrical (cf. Fig 3). While feedforward input $g^{V,FF}$ alone has an excitatory impact on the basal membrane potential, feedback input $h_a^V$ is only effective if it coincides with the interfacing module's gating input $g^{V,B2}$. In addition, the effect of feedback on the excitatory conductance yields a gain enhancement. So, it will up-modulate existing feedforward input (abstractly, i.e., $g^{V,FF} + g^{V,FF} \cdot, g^{V,FB}$; [205]), but

will have no effect if feedforward input is absent. The up-modulation in combination with the inhibitory interaction of locally surrounding neurons implements a biased competition process [205]. For the highest level of the hierarchy, V4, no feedback exists (cf. Table 1). As a result, the apical membrane potential will not receive any input, and thus, will not influence the cell's basal dynamics.

To compute the output rate $r^V$ of the pyramidal cell its basal membrane potential $v_b^V$ is passed through a non-linearity $f^V(x) = min(max(x,0),1)$

$$r^V = f^V\left(v^V\right).$$ (14)

**4.1.3. Thalamic cell model.** Nodes of the interfacing module describe the processing of higher-order thalamic model neurons. The evolution of their membrane potential $v^{HO}$ is governed by the differential equation

$$\tau^{HO}\dot{v}^{HO} = -v^{HO} \cdot g^{HO,Leak} + \left(\beta^{HO} - v^{HO}\right) \cdot g^{HO,Exc}.$$ (15)

The equation is structurally similar to Eqs 9 and 10 with a constant leakage conductance $g^{HO,Leak}$, an excitatory reversal potential $\beta^{HO}$, a time constant $\tau^{HO}$, and an excitatory conductance term $g^{HO,Exc}$. For thalamic cells, this term consists of a summation across multiple inputs $V$ according to

$$g^{HO,Exc} = \sum_V k^{HO,V} \cdot g^{HO,V}, \; V \in \{V1, V2, V4, S\}.$$ (16)

This summation is weighted by constants $k^{HO,V}, V \in \{V1, V2, V4, S\}$ and describes the thalamic evidence accumulation of binding signal information coming from the different neural processors of the visual hierarchy via channels of type $C = B1$ and other sites of the brain $S$ providing a seed of attention. In this case $S$ is the task module providing input to the interfacing module. The accumulated evidence (Eq 16) is mediated by the temporal integration (Eq 15) and then transformed into the cell's output rate $r^{HO}$

$$r^{HO} = f^{HO}\left(v^{HO}\right).$$ (17)

The non-linearity $f^{HO} = 1/(1 + \exp(-(2 \cdot x - 1) \cdot 50))$ is a sigmoid and rather steeply segregates non-binding from binding states by its threshold. This segregation provides a filtering mechanism for up-modulated activity from the visual areas.

## 4.2. Details on experiments

**4.2.1. Simulation details.** All simulations have been implemented with the Julia programming language and have been performed with the same set of model parameters (see Table A in S1 Text for an overview). Only the spatial size and value range of inputs differed for stimuli. Further details about the model and stimuli are provided as supporting information (S1 Text).

To keep the computational cost for simulating the experiments low, steady state iteration has been performed for all dynamical equations instead of a more detailed numerical

integration scheme. We use the following update rules

$$v_b^V \leftarrow \frac{\beta_b^V \cdot g_b^{V,Exc}}{v_b^V \cdot g_b^{V,Leak} + v_b^V \cdot g_b^{V,Exc} + \kappa_b^V \cdot v_b^V \cdot h_b^{V,Inh}}, \quad V \in \{V1, V2, V4\}, \tag{18}$$

$$v_a^V \leftarrow \frac{\beta_a^V \cdot g_a^{V,FB}}{v_a^V \cdot g_a^{V,Leak} + v_a^V \cdot g_a^{V,FB}}, \quad V \in \{V1, V2, V4\}, \tag{19}$$

$$v^{HO} \leftarrow \frac{\beta^{HO} \cdot g^{HO,Exc}}{v^{HO} \cdot g^{HO,Leak} + v^{HO} \cdot g^{HO,Exc}}. \tag{20}$$

See S1 Text for further details on the derivation. For all simulations, all state values have been initialized with zeros.

**4.2.2. Metrics and evaluation schemes.** To evaluate the simulation results, the following metrics and computations have been performed.

The modulation index *MI* has been computed by comparing the neuron's activity between two conditions during each simulation, respectively: the baseline condition at the beginning of the simulation (*base*) and the condition at the end of the simulation (*att*). The *base* condition measures the neuron's activity after a base representation has been formed, while the *att* condition measures the neuron's activity after it has been labeled by attention. The neural activities during these conditions, $\hat{r}_{x_i,y_i,\theta_i,base}^V$, $\hat{r}_{x_i,y_i,\theta_i,att}^V$, served as firing rate estimates to compute the modulation index [16]:

$$MI_{x_i,y_i,\theta_i} = 2 \cdot \frac{\hat{r}_{x_i,y_i,\theta_i,att}^V - \hat{r}_{x_i,y_i,\theta_i,base}^V}{\hat{r}_{x_i,y_i,\theta_i,att}^V + \hat{r}_{x_i,y_i,\theta_i,base}^V} \tag{21}$$

Modulation index computation has been applied to neurons in V1 along the target contour for Fig 4E and 4F, resulting in a total of $n = 2834$ (243 pixel locations along the line with $n_\theta = 12$ neurons per location, respectively, yields 2916 neurons; 82 neurons were excluded for having zero activity in both conditions).

The point in time when the complete curve has been up-modulated by attention has been determined as the modulation onset time of the last neuron position in V1 along the curve's outline (Fig 5). The modulation onset time has been determined for the population's activity summed along the feature dimension $\theta$ to obtain a single onset time per retinotopic position. The onset time was defined as the time step at which neural activity crossed 50% of its maximum across the trial. Linear regression across runs was performed to obtain a fit to tracing times for the parametric variation of target-distractor distances (Fig 5).

The computation for the speed profiles (Fig 6) has been based on the modulation onset time definition described above. To compensate for pixel discretization artifacts and other numerical issues the onset times have been locally smoothed along the contour outline. Smoothing was performed by convolving the onset times with a binomial kernel $\Lambda_s = [1, 4, 6, 4, 1]/16.0$. Then speed values $s_{p_{ij}}$ for positions $p$ along the contour have been computed from pair-wise differences between smoothed onset times of neighboring positions $p_i, p_j$ and subsequently taking the inverse of this onset time difference between both positions, i.e.,

$$s_{p_{ij}} = \frac{1}{\left(t_{p_j} * \Lambda_s\right)(p) - \left(t_{p_i} * \Lambda_s\right)(p)}. \tag{22}$$

In theory the denominator of this equation can become zero or even lead to negative speed estimates if smoothed time estimates for indices $i$ and $j$ do not increase monotonically. In practice no such non-monotonicity has been observed during the simulations.

To compute estimates of the speeds per contour segment (first wide segment, narrow segment, second wide segment; Fig 6) linear regression was performed for the modulation onset times of the neurons per segment. The inverse of the determined slope factor then yielded the speed estimate.

For experiments with complex stimulus geometries (Fig 9) speed estimates have to be compared between different stimulus geometries. Therefore, the abstract (au) speeds (Eq 22) were scaled by a distance estimate $d(p_j, p_i)$ between probe positions along the respective stimulus geometry to obtain comparable units (pixel/step; Fig 9E and Fig A in S2 Text).

## Supporting information

**S1 Text. Further details on methods.** Further details on methods regarding the derivation of the steady-state iteration employed in the simulation, as well as additional details on model parameters and stimulus creation.
(PDF)

**S2 Text. Results on additional stimuli.** Extension of presented results and additional simulations on a broader range of stimuli.
(PDF)

## Acknowledgments

The authors would like to thank Bill Phillips and Sami Mollard for helpful comments on a first preprint version of this manuscript and Pieter R. Roelfsema and Sami Mollard for inspiring discussions in advance to the writing of this manuscript. The authors acknowledge support by the state of Baden-Württemberg through bwHPC.

## Author contributions

**Conceptualization:** Daniel Schmid, Heiko Neumann.

**Data curation:** Daniel Schmid.

**Formal analysis:** Daniel Schmid.

**Funding acquisition:** Heiko Neumann.

**Investigation:** Daniel Schmid.

**Methodology:** Daniel Schmid, Heiko Neumann.

**Project administration:** Daniel Schmid, Heiko Neumann.

**Resources:** Heiko Neumann.

**Software:** Daniel Schmid.

**Supervision:** Heiko Neumann.

**Validation:** Daniel Schmid.

**Visualization:** Daniel Schmid, Heiko Neumann.

**Writing - original draft:** Daniel Schmid, Heiko Neumann.

**Writing - review & editing:** Daniel Schmid, Heiko Neumann.

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
