## [Decision Letter · Decision Letter 0]

26 Mar 2024

Dear Mr Schmid,

Thank you very much for submitting your manuscript "Thalamo-cortical interaction for incremental binding in mental contour-tracing" for consideration at PLOS Computational Biology.

As with all papers reviewed by the journal, your manuscript was reviewed by members of the editorial board and by several independent reviewers. In light of the reviews (below this email), we would like to invite the resubmission of a significantly-revised version that takes into account the reviewers' comments.

Although the reviewers were generally positive, they also raised some important concerns, including the consistency of the model with neurobiology, the scalability of the model to more complex imaging and other stimulus configurations, and the clarity of the explanation.

We cannot make any decision about publication until we have seen the revised manuscript and your response to the reviewers' comments. Your revised manuscript is also likely to be sent to reviewers for further evaluation.

Sincerely,

Ariel Zylberberg, Ph.D.

Guest Editor

PLOS Computational Biology

Lyle Graham

Section Editor

PLOS Computational Biology

Reviewer #1: The manuscript describes a new neurodynamical model of incremental grouping and object-based attention. The model is based on biophysically realistic components and describes how interactions between the visual cortex and thalamus achieve binding of visual elements into a distinct representation of target and distractor curves by labeling target curve with enhanced firing rate. The model was successfully tested on several stimulus configurations similar to those used in empirical studies. The authors also offer a complexity analysis which suggests that the proposed model is an efficient implementation of incremental grouping. The manuscript offers a new perspective on the neural mechanisms that underlie interaction between vision and attention and it represents significant advancement over existing theoretical approaches. Thus, I recommend the publication of this work provided that the authors revise and expand it along the points listed below.

1. Stimulus configurations

The model was tested on two stimulus configurations that are important but not sufficient to fully capture the complexity of curve tracing. What is needed is a more comprehensive test to assess the model’s capabilities and limitations. I suggest further testing along the following lines:

- how would the model behave in response to a configuration where two curves meet like this >< ? If, for example, tracing starts on the left curve it should stay on the left without crossing to the right side. Similar configurations consisting of >< and X have been used by Houtkamp et al. (2003, Perception & Psychophysics) so the model could be tested on these inputs. In general, how would the model perform tracing along the acute angles? I suspect that this would require more complex connectivity scheme and maybe even introduction of more units dedicated to the representation of junctions as shown by the model of Marić and Domijan (2019, Neural Networks; Domijan & Marić, 2022, Vision Research).

- how would the model behave in response to curves of various degrees of curvatures? Behavioral studies showed that tracing become slower as the contour curvature increases (Jolicoeur et al., 1991, JEP: HPP). If it is too difficult to construct such stimuli, this topic should be discussed in more detail in the Discussion.

- could the model perform tracing along the broken curves? This could be tested by inserting small gaps into the target and distractor curves. This would greatly enhance model’s explanatory power because experimental studies showed that tracing along the broken curves proceeds at approximately the same speed as tracing along complete curves (Houtkamp & Roelfsema, 2010, JEP: HPP).

2. Biological considerations

- Authors attribute curve tracing to interaction between visual cortex and thalamus. However, it is known that thalamus exhibits oscillations, including periods of bursting. Furthermore, recent studies suggest that thalamus (pulvinar) generates slow theta band oscillations that rhythmically modulate cortex (Fiebelkorn et al., 2019, Nature Communications; Fiebelkorn & Kastner, 2019, Trends in Cognitive Sciences). How would such oscillations affect curve tracing in the model? Authors touched upon on this question in the Discussion but this should be expanded with detailed description of how would the model behave in the presence of such oscillations.

- Authors noted that their model do not need separate classes of units as other models do. In particular, several models rest on the division of labor between so called A- and N-units (attentional and non-attentional units, respectively). This design choice is motivated by single-units recordings showing that some but not all neurons in monkey V1 are subject to attentional modulations (Pooresmaeili et al., 2010, Journal of Neuroscience). Moreover, the cited study reported that approximately half of the recorded units are unaffected by attention. This should be acknowledged when comparing different models.

- If I understood this correctly, spatial spread of feedforward and feedback projections between V1 and V2 and between V2 and V4 are of equal size which is not anatomically correct because spread of feedback projections are typically much larger than that of feedforward projections (Angelucci et al., 2002, Journal of Neuroscience). How would an increase in spatial spread of feedback projections affect model’s behavior? This could be discussed in relation to the question of how model parameters were chosen as noted in the point below.

3. Model description

- The model is defined by a set of differential equations, but simulations are performed on their steady-state approximation (p. 35). I think it would be more transparent to explicitly report how steady-state approximation is derived and to describe exact equations that are used in simulations. This can be put in a separate subsection of Appendix.

- Authors shortly described how they set the model parameters (p. 35). However, this section could be expanded a bit by giving more information regarding which parameters are critical for model performance and which are set to some generic values without much effect on model’s behavior. For example, by reading from Table 2, it seems that spatial extent of kernel from HO thalamus to V1, V2 and V4 is critical to generate tracing with variable speeds. On the other hand, envelope standard deviation for feedforward and feedback projections seems to be of less importance as it was set to equal value of 2.0 (this requires more justification given the fact that feedback projections are usually wider that corresponding feedforward projections as noted above). Such observations should be stated explicitly.

- In all differential equations, the dot above variable on the right-hand side of equation is missing.

Reviewer #2: Schmid and Neumann present a description of their implementation of the ‘growth-cone model’, which has previously been proposed by Roelfsema and his colleagues. The model proposes that the perceptual grouping of objects is achieved in a time-consuming, incremental process. The speed of processing depends on the size of the response fields that can contribute to the grouping: detailed features need to be resolved at a high resolution (e.g. V1), whereas larger segments can be resolved at a lower resolution if there is no interference from other objects nearby in the visual scene.

I have three main issues that I hope the authors will be able to resolve.

1) Scaling response field size with eccentricity

The model does not seem to take into account that receptive field size scales with eccentricity. Does this mean that the model only works for stimuli of a limited, small size? If so, I am concerned that the model can only operate in a very limited set of conditions, and is not relevant for more realistic visual scenes. Alternatively, does this mean that we should not think of the levels in a hierarchical way (such as labeled, V1, V2, and V4 in Figure 2), but rather a conceptual description of having various sizes at various levels for different eccentricities? If so, I am concerned that the link between the model and the biology (e.g. labeling areas as V1, V2, etc) is too tight, and not relevant.

2) Unclear role of the interface module

The original proposal for the growth-cone model incorporates the idea of feedback within the visual hierarchy (V1, V2, V4). Feedback to V1 from areas with larger receptive fields should help increase the speed of grouping. Why is there a need for an ‘interface module’ and why should lesioning its connection to visual areas decrease the speed of grouping? Shouldn’t the speed still increase on the wider segments due to feedback from V2/4 (beyond the small difference in speeds sown in Figure 5)? I wonder if at all there is a need to separate out the interfacing module and the task module. Since the weights in Equation 16 already incorporate the “seed of attention”, why does it have to come from a different module and not be self-contained?

3) Loose link between model and biology

The link between the computational model and the biological mechanism seems very loose and not well-motivated. Why is the ‘Interfacing module’ similar to Higher-order thalamus? What is the ‘apical integration site’ vs ‘basal integration site’ distinction based on? These parallels are presented as if they are well-known facts, which I believe is misleading. The study certainly doesn’t provide additional evidence beyond what is already in the literature that thalamo-cortical interactions are involved in incremental binding, as suggested by the title of the manuscript. The authors have 3 options to solve this: 1) they could better support the link between the model and the biology with citations to the experimental literature and discuss alternatives, or 2) only mention the parallels that *may* exist between the model and biology in the discussion, or 3) make more clear that they are trying to make testable predictions that remain to be tested. Or a combination of all these.

Furthermore, I have a few comments that I hope will help the authors further improve the manuscript.

The abstract, author summary, and introduction suggest that object-based attention and serial processing is only needed for ‘complicated objects’. The curve-tracing stimuli have in the past sometimes be criticized for being overly complex or artificial (i.e., it is like visually untangling a bunch of electrical cords). On the other hand, natural stimuli could be considered more complex as they have more features and complex shapes. The growth-cone hypothesis has been proposed and tested experimentally for both situations, with curve tracing being covered in reference 10, and simple shapes and other complex images covered in reference 11. Can the authors please clarify what they mean by ‘complicated objects’? The stimuli used in the current study are simple lines, so the manuscript would benefit from a discussion of how the results may generalize to other stimuli.

The authors are aware of some of the relevant literature on incremental grouping in natural scenes (their reference 11 and 91), but suggest in their introduction that incremental grouping would only be needed when “Ambiguities […] can’t be resolved by parallel grouping” [Page 3]. A more accurate description of previous work would help clarify the generality of the experimental findings, which covers curves/lines as well as 2D shapes and natural images. A distinction between recognition (largely believed to rely on feedforward mechanisms), and grouping (assigning elements to an object, believed to rely on the incremental mechanism studied here) may help here.

The experimental finding described in figure 1a are all from macaque monkeys. The primate visual system has some unique features that are different from other species (e.g. foveal vision, a visual hierarchy with increasing receptive field sizes and tuning complexity, receptive field sizes that scale with eccentricity, clustered/columnar functional tuning, etc). Many of these features are relevant for the presented model. It would therefore be useful to the reader to mention that the cited experimental work is mostly about the primate visual system (human and monkey). This is especially relevant in the figure caption of figure 1a, but also in some other places, especially in the introduction and discussion.

At what scale of receptive field size does the Interfacing module operate? Is it the same spatial resolution as the V1 layer? If so, why is there a need for an interfacing module? Would up-regulating activity in the V1 module be sufficient? If the interfacing module operates at a coarser scale, what would happen if the two endpoints of the curves are close to each other? Could the model confuse the curves at this level?

The figure caption for Figure 3 could be improved by an explanation of what the reader should get from panels e and f. When is baseline activity measured? How is its relation to up-modulated activity interpreted? What is the n exactly in n=2834? Does it group across all modules? How is modulation index computed? A reference to a specific place in the method section would be helpful in finding the relevant information.

The authors make a real effort to ensure that the reader can understand the paper without the need to go through all the math, which I’m sure many readers will greatly appreciate. However, on page 11, the measure of ‘neurons/step’ could be better explained to give some insights into what this measure means.

Please label the axis in Fib. 5a, v1 activity over time. How many steps are on the x-axis, and how many neurons on the y-axis?

In Fig. 5c, it is unclear why the deviations from the fitted lines are so large. Please clarify this. Would this go away when averaging across multiple iterations? If so, why was this not done? If not, why not?

It is unclear what the ablation experiment that disconnects the higher visual areas (e.g. V4) to lower areas (V2) adds to our understanding of the growth-cone model. In reference 11, the authors compare the grouping speed for the growth-cone model to the ‘pixel-by-pixel model’ (McCormick & Jolicoeur, 1994; Jolicoeur & Ingleton, 1991), which, similar to disregarding feedback in the ablation study, sets the grouping speed to a fixed number for each step, regardless of scale/contours. What does the current study add to this? How would the results change if the scaling of receptive field size with eccentricity would be taken into account?

The introduction to graph network and the steps to potentially optimize such a network with NProc units have been thoroughly explored by the authors. However, it is unclear why one would choose a graph network among other alternatives. A connection can be made between the complexity insights and the pyramidal cell model and one of the goals of this paper seems to be bridging the levels of Marr. However, it would be helpful to see (1) A rationale behind the choice of graph network for this particular model and (2) a few testable predictions from the model simulations that hint how future studies can explore the legitimacy of the extensive optimal complexity exercise that the authors went through (3) How exactly were the complexity optimizations incorporated in the simulations? (4) Figure 7 is very dense and would benefit from more explanation, for example, is 7c a simulation based figure or a concept figure?

The manuscript/introduction would benefit from a better coverage of previous work. For example: In the first paragraph of introduction where incremental grouping is introduced as “Incremental grouping iteratively and dynamically evaluates compatibility of previously grouped elements with their neighbors” only one paper is cited.

The writing is sometimes confusing or inexact, and the manuscript would benefit from clarifying some concepts and more accurate use of terms. A few examples:

⁃ “Visual cortex is able to solve a variety of tasks” [page 3], followed by tasks such as “visual comparison judgements”. It is generally accepted that more than just the visual system (which is more than visual cortex) would be involved in making perceptual judgements. Using ‘visual system’ or ‘brain’ may therefore be more accurate.

⁃ “Object based attention is one such task” [page 3]. Attention could be called a cognitive function, or many other things, but it is not a task.

Minor: The introduction aims to link the work to the three levels of Marr (their reference 16). If this is indeed important, it would help to have a conceptual figure indicating which parts of the paper caters to the different levels of analysis and how they are connected.

Minor: it would be useful to reference specific figure panels and sections of the methods in the relevant parts of the result sections.

Reviewer #3: A cortico-thalamic model of multi-scale incremental feature binding

Our retinae sample the images in our eyes discretely, conveying a million local measurements through the optic nerve to our brains. Given this piecemeal mess of signals, our brains infer the structure of the scene, giving us an almost instant sense of the geometry of the environment and of the objects and their relationships.

We see the world in terms of objects. But how our visual system defines what an object is and how it represents objects is not well understood. Two key properties thought to define what an object is in philosophy and psychology are spatiotemporal continuity and cohesion (Scholl 2007). An object can be thought of as a constellation of connected parts, such that if we were to pull on one part, the other parts would follow along, while other objects might stay put. Because the parts cohere, the region of spacetime that corresponds to an object is continuous. The decomposition of the scene into potentially movable objects is a key abstraction that enables us to perceive, not just the structure and motion of our surroundings, but also the proclivities of the objects (what might drop, collapse, or collide) and their affordances (what might be pushed, moved, taken, used as a tool, or eaten).

An important computational problem our visual system must solve, therefore, is to infer what pieces of a retinal image belong to a single object. This problem has been amply studied in humans and nonhuman primates using behavioral experiments and measurements of neural activity. A particular simplified task that has enabled highly controlled experiments is mental line tracing. A human subject or macaque fixating on a central cross is presented with a display of multiple curvy lines, one of which begins at the fixation point. The task is to judge whether a peripheral red dot is on that line or on another line (called a distractor). Behavioral experiments show that the task is easy to the extent that the target line is short or isolated from any distractors. Adding distractor lines in the vicinity of the target line to clutter up the scene and making the target line long and curvy makes the task more difficult. If the target snakes its way through complex clutter closeby, it is no longer instantly obvious where it leads and attention and time are required to judge whether the red dot is on the target or on a distractor line.

Our reaction time is longer when the red dot is farther from fixation along the target line. This suggests that the cognitive process required to make the judgment involves tracing the line with a sequential algorithm, even when fixation is maintained at the central cross. However, the reaction time is not in general linear in the distance, measured along the line, between the fixation point and the dot, as would be predicted by sequential tracing of the line at constant speed. Instead, the speed of tracing is variable depending on the presence of distracting lines in the vicinity of the current location of the tracing process along the target line. Tracing proceeds more slowly when there are distracting lines close by and more quickly when the distracting lines are far away.

The hypothesis that the primate visual system traces the line sequentially from the fixation point is supported by seminal electrophysiological experiments by Pieter Roelfsema and colleagues, which have shown that neurons in early visual cortex that represent particular pieces of the line emanating from the fixation point are upregulated in sequence, consistent with a sequential tracing process. This sequential upregulation of activity of neurons representing progressively more distal portions of the line is often interpreted as the neural correlate of attention spreading from fixation along the attended line during task performance.

The variation in speed of the tracing process can be explained by the attentional growth-cone hypothesis (Pooresmaeili & Roelfsema 2014) which posits that attention spreads not only in the primary visual cortex but also at higher levels of cortical representation. This hypothesis can explain the variation in tracing speed: At higher levels of cortical visual representation, neurons have larger receptive fields and offer a coarser-scale summary of the image, enabling the tracing to proceed at greater speed along the line in the image. In the absence of distractors, tracing can proceed quickly at a high-level of representation. However, in the presence of distractors, the higher-level representations may not be able to resolve the scene at a sufficient grain, and tracing must proceed more slowly in lower-level representations.

Higher-level neurons are more likely to suffer from interference from distractor lines within their larger receptive fields. If a distractor line is present in a neuron’s receptive field, the neuron may not respond as strongly to the line being traced, effectively blocking the path for sequential tracing in the high-level representation. However, tracing can continue – more slowly – at lower levels, where receptive fields are small enough to discern the line without interference.

Detail from Fig. 6 in Pooresmaeili et al. (2014) illustrating the single-scale tracing model (left) and the growth-cone model (right), in which the attentional label is propagated from the fixation point (small red dot) at all levels of representation where receptive fields (circles) do not overlap with the distractor curve. Tracing proceeds rapidly at coarse scales (orange, blue) where the target line is far from the distractor and slowly at fine scales (yellow, green) where the target curve comes close to the distractor.

Now Schmid & Neumann (pp2024) offer a brain-computational model explaining in detail how this multiscale algorithm for attentional selection of the line emanating from the fixation point might be implemented in the primate brain. They describe a mechanistic model and demonstrate by simulation that it can explain how mental line tracing might be implemented in the primate brain.

Pyramidal neurons at multiple levels of the visual hierarchy (corresponding to cortical areas V1, V2, V4) detect local oriented line segments on the basis of the bottom-up signals arriving at their basal dendritic integration sites. These line segments are pieces of the target and distractor lines, represented in each area at a different scale of representation. The pyramidal neurons also receive lateral and top-down input providing contextual information at their apical dendritic integration sites, enabling them to sense whether the line segment they are representing is part of a longer continuous line.

The attentional “label” indicating that a neuron represents a piece of the target line is encoded by an upregulation of the activity of the pyramidal neurons, consistent with neural recording results from Roelfsema and colleagues (1998). The upregulation of activity, i.e. the attention label, can spread laterally within a single area such as V1. Connectivity between neurons representing approximately collinear line segments implements an inductive bias that favors interpretations conforming to the Gestalt principle of good continuation. However, the upregulation will spread only to pyramidal neurons that (1) are activated by the stimulus, (2) receive contextual input from pyramidal neurons representing approximately collinear line segments, and (3) receive thalamic input indicating the local presence of the attentional marker.

Each step of propagation is conditioned on the conjunction of these three criteria. The neural computations could be implemented exploiting the intracellular dynamics in layer-5 pyramidal neurons, where dendritic inputs entering at apical integration sites cannot drive a response by themselves but can modulate responses to inputs entering at basal integration sites. An influential theory suggests that contextual inputs arriving at the apical dendritic integration sites modulate the response to bottom-up stimulus inputs arriving at the basal dendritic integration sites (Larkum 2013, BrainInspired podcast). Schmid and Neumann’s model further posits that the apical inputs are gated by thalamic inputs (Halassa & Kastner 2017), implementing a test of the third criterion for propagation of the attentional label.

The attentional label is propagated locally from already labeled pyramidal neurons to pyramidal neurons at all levels of the visual hierarchy that represent closeby line segments sufficiently aligned in orientation to be consistent with their being part of the target line. To enable the coarser-scale representations in higher cortical areas to speed the process, neurons representing the same patch of the visual field at different scales are connected through thalamocortical loops. Through the thalamus, each level is connected to all other levels, enabling label propagation to bypass the stages of the hierarchy. The thalamic component (possibly in the pulvinar region of the visual thalamus) represents a map of the labeled locations, but not detailed orientation information.

Imagine a mechanical analogy, in which tube elements represent local segments of the lines. The stimulus-driven bottom-up signals align the orientations of the tube elements with the orientations of the line segments they represent, so the tube elements turn to form long continuous tunnels depicting the lines. A viscous liquid is injected into the tube element representing the fixation point and spreads. Adjacent tube elements need to be aligned for the liquid to flow from one into the other. In addition, there are valves between the tube elements, which open only in the presence of thalamic input. Importantly, the viscous liquid can flow not only at the V1 level of representation, where the tube elements represent tiny pieces of the lines and the viscous liquid needs to flow through many elements to reach the end of the line. Rather, the liquid can also take shortcuts through higher-level representations, where long stretches of the line are represented by few tube elements. This enables the liquid to reach the end of the line much more quickly – to the extent that there are stretches sufficiently isolated from the distractors for coarse-scale representation at higher levels of the hierarchy.

Since the information about (1) the presence of oriented line segments, (2) their compatibility according to the Gestalt principle of good continuation, and (3) the attentional label are all available in the cortical hierarchy, a growth-cone algorithm could be implemented without thalamocortical loops. However, Schmid and Neumann argue that the non-orientation-specific thalamic representation reduces the complexity of the circuit. Fewer connections are required by decomposing the question “Are there upregulated compatible signals in the neighborhood?” into two simpler questions: “Are there compatible signals in the neighborhood?” (answered by cortex) and “Are there upregulated signals in the neighborhood?” (answered by the thalamic input). Because there could be compatible signals in the neighborhood that are not upregulated, and upregulated signals that are not compatible, yeses to both questions of the decomposition do not in general imply a yes to the original question. However, if we assume that there is only one line segment per location, then two yeses do imply a yes to the original question.

Schmid and Neumann argue that thalamic label map enables a simpler circuit that works in the simulations presented, even tracing a line as it crosses another line without spillover. We wonder if, in addition to requiring fewer connections, the thalamic label map might have functional advantages in the context of a system that must be able to perform not just line tracing but many other binding tasks, where the thalamus might have the same role, but the priors defining compatibility could differ.

Why is this model important? Line tracing is a type of computational problem that is prototypical of vision and yet challenging for both of our favorite modes of thinking about visual computations: deep feedforward neural networks and probabilistic inference. These two approaches (discriminative and generative to a first approximation) form diametrically opposed corners in a vast space of visual algorithms that has only begun to be explored (Peters et al. pp2023). Line tracing is a simple example of a visual cognition task that can be rendered intractable for both approaches by making the line snaking its way through the clutter sufficiently long and the clutter sufficiently close and confusing. Feedforward deep neural networks have trouble with this kind of problem because there are no hints in the local texture revealing the long-range connectivity of the lines. The combinatorics creates too rich a space of possible curves to represent with a hierarchy of features in a neural network. Although any recurrent computation (including the model of Schmid and Neumann and a recent line tracing model from Linsley & Serre, 2019) can be unfolded into a feedforward computational graph, the feedforward network would have to be very deep, and its parameters might be hard to learn without the inductive bias that iterating the same local propagation rule is the solution to the puzzle (van Bergen & Kriegeskorte 2020). From a probabilistic inference perspective, similarly, the problem is likely intractable in its general form because of the exponential number of possible groupings we would need to compute a posterior distribution over.

By assuming that we can be certain about the way things connect locally, we can avoid having to maintain a probability distribution over all possible line continuations from the fixation point. Binarizing the probabilities turns the problem into a region growing (or graph search) problem requiring a sequential procedure, because later steps depend on the result of earlier steps.

Schmid and Neumann’s paper describes how the previously proposed growth-cone algorithm, which solves an important computational challenge at the heart of visual cognition (Roelfsema 2006), might be implemented in the primate brain. The paper seriously engages both the neuroscience (at least at a qualitative level) and the computational problem, and it connects the two. The authors simulate the model and demonstrate its predictions of the key behavioral and neurophysiological results from the literature. They use model-ablation experiments to establish the necessity of different components. They also describe the model at a more abstract level: reducing the operations to sequential logical operations and systematically considering different possible implementations in a circuit and their costs in terms of connections. This resource-cost perspective deepens our understanding of the algorithm and reveals that the proposed model is attractive not only for its consistency with neuroanatomical, neurophysiological, and behavioral data, but also for the efficiency of implementation in a physical network.

Strengths

- Offers a candidate explanation for how an important cognitive function might be implemented in the primate brain, using an algorithm that combines parallel computation, hierarchical abstraction, and sequential inference.

- Motivated by a large body of experimental evidence from neurophysiological and behavioral experiments, the model is consistent with primate neuroanatomy, neural connectivity, neurophysiology, and subcellular dynamics in multi-compartment pyramidal neurons.

- Describes a class of related algorithms and network implementations at an abstract level, providing a deeper understanding of alternative possible neural mechanisms that could perform this cognitive function and their network complexity.

Weaknesses

- The model operates on a toy version of the task, using abstracted stimuli with few orientations and predefined Gabor filter banks as model representations, rather than more general visual representations learned from natural images. An important question is to what extent the algorithm will be able to perform visual tasks on natural images. Given the complexity of the paper as is, this question should be considered beyond the scope, but related work connecting these ideas to computer vision could be discussed in more detail.

Major suggestions

(1) Illustrate the computational mechanism and operation of the model more intuitively. In Fig. 1b, colors code for the level of representations. It would therefore be better to not use green to code for the selection tag. Thicker black contours or some other non-color marker could be used. It is also hard to see that the no-interference and the interference cases have different stimuli. Only the bottom panels with the stimuli show a slight difference. The top panels should be distinct as well since different neurons would be driven by the two stimuli. Alternatively, you could consider using only one stimulus, where the distractor distance variation is quite pronounced, but showing time frames to illustrate the variation of the speed of the progression of attentional tagging.

(2) Discuss challenges in scaling and calibrating the model for application to natural continuous curves. The stimuli analyzed have only a few orientations with sudden transitions from one to the other. Would the model as implemented also work for continuous curves such as those used in the neurophysiological and behavioral experiments or would a finer tiling of orientations be required? Under what conditions would attention spill over to nearby distractor curves? It would be good to elaborate on the roles of surround suppression, inhibition among detectors, and the excitation/inhibition balance.

(3) Discuss challenges in scaling the model to computer vision tasks on natural images. To be viable as brain-computational theories, models ultimately need to scale to natural tasks. Please address the challenges of extending the model for application to natural images and computer-vision tasks. This will likely require the representations to be learned through backpropagation. The cited complementary work by Linsley and Serre on the pathfinder task using horizontal gated recurrent units and incremental segmentation for computer vision is relevant here and deserves to be elaborated on in the Discussion. In particular, do the growth-cone model and your modeling results suggest an alternative neural network architecture for learning incremental binding operations?

Minor suggestions

(1) Please make sure that the methods section contains all the details of the model architecture needed for replication of the work. Much of the math is described well. But some additional technical details on maps and connectivity may be needed. What are the sizes of the maps? What do they look like for a given input? Do they appear like association fields? What is the excitatory and inhibitory connectivity as a function of spatial locations and orientations of the source and target unit?

(2) Discuss how the model relates to the results of Chen et al. (2014) who described the interplay between V1 and V4 during incremental contour integration on the basis of simultaneous recordings in monkeys.

(3) Although the paper is well-written and clear, the English is a bit rocky throughout with many grammatical errors and some typos. These could be fixed using a proofreader or suitable software.

Some examples…

Fig. 1 legend: “distance among the line” -> “distance along the line”

“which triggers the execution of the execution of the incremental binding process”

->“which triggers the execution of the incremental binding process”

“only one proportion” -> “only a portion”

There are many little bugs like that throughout.

– Nikolaus Kriegeskorte & Hossein Adeli

**Have the authors made all data and (if applicable) computational code underlying the findings in their manuscript fully available?**

Reviewer #1: Yes

Reviewer #2: Yes

Reviewer #3: **No: **A link to a github repository is included, but it appears to be empty.

PLOS authors have the option to publish the peer review history of their article (what does this mean?). If published, this will include your full peer review and any attached files.

Reviewer #1: No

Reviewer #2: No

Reviewer #3: **Yes: **Nikolaus Kriegeskorte (with Hossein Adeli)
---

## [Decision Letter · Decision Letter 1]

8 Oct 2024

Dear Mr Schmid,

Thank you very much for submitting your manuscript "A model of thalamo-cortical interaction for incremental binding in mental contour-tracing" for consideration at PLOS Computational Biology. Your manuscript was reviewed by members of the editorial board and by the same three reviewers that reviewed the original submission.

The three reviewers were very positive about the revised manuscript. Based on the reviews, we are likely to accept your manuscript for publication. There is one remaining comment from reviewer 2 regarding the increase in receptive field size at larger eccentricities that we would like you to address. We may send your response to this reviewer for further evaluation before we can approve your submission for publication.

Sincerely,

Ariel Zylberberg, Ph.D.

Guest Editor

PLOS Computational Biology

Lyle Graham

Section Editor

PLOS Computational Biology

The three reviewers were very positive about the revised manuscript. There is one remaining comment from reviewer 2 regarding the increase in receptive field size at larger eccentricities that we would like you to address. We may send your response to the reviewer for further evaluation before we can approve your submission for publication.

Reviewer's Responses to Questions

**Comments to the Authors:**

Reviewer #1: In the revised manuscript, authors adequately addressed all the concerns raised by the reviewers so it can be accepted for publication in the current form. I appreciate great effort that the authors invested in extending their work with additional simulations and in clarifying conceptual issues. Just one last point is that authors should make their code publicly available since, at the moment, the link in the footnote on page 45 points to an empty page.

Reviewer #2: I believe the authors have improved their manuscript. My main comment about this revision is regarding the scaling of RF sizes by eccentricity as it seems that the authors may have misunderstood that part of my initial review. In my original review, I suggested that the authors take into account the effect of increased receptive field size for larger eccentricities. With larger receptive field sizes at larger eccentricities, the speed of grouping should be faster for larger eccentricities, and the effect of lesioning be more severe for the grouping processes at these image locations. The original paper ignored this fact, a fact that is well known from electrophysiology experiments. The authors are right that the effect of eccentricity on RF size in V1 is rather small, but it’s more pronounced in higher regions, which are more relevant in this case. A foveal response field in V4 is very small, but at just a few degrees away from the fovea its rather large. I appreciate that the authors have attempted to address this, however, their approach of blurring the images (Supplement S2) does not seem like a valuable addition to their paper. Blurring the input image could lead to errors in grouping, which, indeed, the authors find: grouping may erroneously switch to the distractor, or stop altogether (page 6 in Supplement). I do not understand the logic: How can blurring the input image approximate the results that one would get when larger grouping speeds are encouraged at larger eccentricities due to larger receptive fields? Blurring the input does not simulate the effect that we may see that is caused by large V4 RFs providing feedback to large groups of cells in V1. I would encourage the authors to consider testing the model in such a way that RF size is taken into account (V2 and V4 are more important than V1, as they provide the feedback). Would the results of the lesion studies be different?

Reviewer #3: This is a thorough and comprehensive revision making a good paper great.

The authors have addressed all my substantial concerns.

The paper is now ready for publication in PLoS Computational Biology.

**Have the authors made all data and (if applicable) computational code underlying the findings in their manuscript fully available?**

Reviewer #1: **No: **The link to online code in the manuscript points to the empty page.

Reviewer #2: Yes

Reviewer #3: **No: **Not yet, it appears, but they say that it will be made available in the github repository upon publication.

PLOS authors have the option to publish the peer review history of their article (what does this mean?). If published, this will include your full peer review and any attached files.

Reviewer #1: **Yes: **Dražen Domijan

Reviewer #2: No

Reviewer #3: **Yes: **Nikolaus Kriegeskorte

Figure Files:

Data Requirements:

Reproducibility:

References:

---

## [Editor Report · Decision Letter 2]

29 Jan 2025

Dear Mr Schmid,

We are pleased to inform you that your manuscript 'A model of thalamo-cortical interaction for incremental binding in mental contour-tracing' has been provisionally accepted for publication in PLOS Computational Biology.

Best regards,

Ariel Zylberberg, Ph.D.

Guest Editor

PLOS Computational Biology

Lyle Graham

Section Editor

PLOS Computational Biology

---

## [Editor Report · Acceptance letter]

PCOMPBIOL-D-23-02106R2

A model of thalamo-cortical interaction for incremental binding in mental contour-tracing

Dear Dr Schmid,

I am pleased to inform you that your manuscript has been formally accepted for publication in PLOS Computational Biology. Your manuscript is now with our production department and you will be notified of the publication date in due course.

With kind regards,

Anita Estes
